# Molecular Alterations of the Endocannabinoid System in Psychiatric Disorders

**DOI:** 10.3390/ijms23094764

**Published:** 2022-04-26

**Authors:** Daniela Navarro, Ani Gasparyan, Francisco Navarrete, Abraham B. Torregrosa, Gabriel Rubio, Marta Marín-Mayor, Gabriela B. Acosta, Maria Salud Garcia-Gutiérrez, Jorge Manzanares

**Affiliations:** 1Instituto de Neurociencias, Universidad Miguel Hernández-CSIC, Avda. de Ramón y Cajal s/n, San Juan de Alicante, 03550 Alicante, Spain; dnavarro@umh.es (D.N.); agasparyan@umh.es (A.G.); fnavarrete@umh.es (F.N.); a.bailen@umh.es (A.B.T.); maria.ggutierrez@umh.es (M.S.G.-G.); 2Redes de Investigación Cooperativa Orientada a Resultados en Salud (RICORS), Red de Investigación en Atención Primaria de Adicciones (RIAPAd), Instituto de Salud Carlos III, MICINN and FEDER, 28029 Madrid, Spain; grubiovalladolid@gmail.com; 3Instituto de Investigación Sanitaria y Biomédica de Alicante (ISABIAL), 03010 Alicante, Spain; 4Instituto de Investigación (i+12), Hospital Universitario 12 de Octubre, 28041 Madrid, Spain; marta.marin@salud.madrid.org; 5Department of Psychiatry, Complutense University of Madrid, 28040 Madrid, Spain; 6Instituto de Neurociencias Cognitiva y Traslacional (INCYT), CONICET, INECO, Universidad Favaloro, Ciudad Autónoma de Buenos Aires C1079ABE, Argentina; gabypentin38@gmail.com

**Keywords:** endocannabinoid system, psychiatric disorders, molecular alteration, method

## Abstract

The therapeutic benefits of the current medications for patients with psychiatric disorders contrast with a great variety of adverse effects. The endocannabinoid system (ECS) components have gained high interest as potential new targets for treating psychiatry diseases because of their neuromodulator role, which is essential to understanding the regulation of many brain functions. This article reviewed the molecular alterations in ECS occurring in different psychiatric conditions. The methods used to identify alterations in the ECS were also described. We used a translational approach. The animal models reproducing some behavioral and/or neurochemical aspects of psychiatric disorders and the molecular alterations in clinical studies in *post-mortem* brain tissue or peripheral tissues were analyzed. This article reviewed the most relevant ECS changes in prevalent psychiatric diseases such as mood disorders, schizophrenia, autism, attentional deficit, eating disorders (ED), and addiction. The review concludes that clinical research studies are urgently needed for two different purposes: (1) To identify alterations of the ECS components potentially useful as new biomarkers relating to a specific disease or condition, and (2) to design new therapeutic targets based on the specific alterations found to improve the pharmacological treatment in psychiatry.

## 1. Introduction

Cannabis Sativa L. plant (cannabis) has been used for medicinal and nonmedicinal purposes for centuries. Cannabis has more than 120 cannabinoid compounds, with Δ^9^-tetrahydrocannabinol (THC) and cannabidiol (CBD), the most abundant and widely used [1]. Cannabis consumption has increased in the last decades due to arbitrary use to treat different diseases such as chronic pain and psychiatric disorders [2,3,4] without the necessary information of double-blind placebo (PLB) clinical trials and approval of the corresponding National or International Health Agencies. Synthetic cannabinoids such as Nabilone have been used to treat symptoms present in psychiatric disorders [5]. Many adverse effects of cannabis use have been described such as an association between cannabis use and the development of psychosis in patients with an immature central nervous system (CNS) [6,7].

On the other hand, it is worth highlighting that cannabinoids and their derivatives present psychoactive effects such as euphoria, paranoia or anxiety, disturbance of the perception of time and space, fatigue, problems with motor coordination, confusion, and impaired concentration [8]. Impaired memory and concentration and a tendency to depressive behavior, paranoia, delusions, hallucinations, anxiety, panic attacks, agitation, nausea, vomiting, seizures, and dizziness are also adverse effects of cannabis use [9,10]. These adverse effects, mainly produced by cannabinoid receptor 1 (CB1r) activation, must be considered in treating psychiatric disorders. It is highly relevant to mention that the responses to cannabinoid medications may depend, at least in significant part, on the patient’s ECS activity, the proportion, and the dosage of phytocannabinoids and synthetic derivatives used. Moreover, it is crucial to consider the pharmacological interactions between cannabinoid compounds and conventional drugs to treat psychiatric disorders. 

The ECS has an essential role in modulating the brain’s rewards functions, emotion, appetite, motivational behavior, and cognitive function [11]. ECS components (receptors, ligands, synthesizing and degrading enzymes) have become highly interesting because of their neuromodulator role in glutamatergic, γ-aminobutyric acid (GABA)-ergic (GABAergic), serotonergic, and dopaminergic mechanisms in the CNS. CB1rs are present on inhibitory, GABAergic interneurons, and in excitatory glutamatergic terminals, although to a lesser extent [12], and on dopaminergic neurons, playing a specific role in different emotional behaviors affected in psychiatric disorders [13]. The modulation of cannabinoid receptor 2 (CB2r) induces anxiolytic [14,15] and antidepressant activity [16,17], which may be of great interest in managing a mental illness. N-arachidonylethanolamine (anandamide, AEA) and 2-arachidonoylglycerol (2-AG) may have several beneficial effects, such as modulating anxiety and depression, and pain. Indeed, in animal studies, increasing endogenous AEA or 2-AG pharmacologically produces anti-anxiety effects under stressful conditions [18]. 

The ECS is also strongly involved in establishing neuronal diversity and synaptic connectivity in the developing brain, where cellular proliferation, migration, and maturation are essential to achieving normal corticogenesis [19]. Cannabinoid receptors and their endogenous ligands AEA and 2-AG are present in the rat brain as early as gestational days 11–14 [20,21], while in humans, CB1r was found from week 14 of gestation [22,23]. Therefore, considering the ECS involvement in brain development and genetic and epigenetic variations affecting its components, it could be associated with abnormal brain development. Therefore, pre- and postnatal exposure to different environmental agents, such as stress or toxic substances can induce other effects on the ECS, changing the basal tone of this system and increasing the vulnerability to developing psychiatric illnesses later in life [6,7,24]. Abnormal CNS development can show disrupted brain size and morphology associated with developmental delay and intellectual disability. Some psychiatric disorders present important neurodevelopmental bases such as autism spectrum or attention deficit and hyperactivity disorders. Moreover, in different psychiatric and neurodevelopmental disorders, several molecular changes in ECS components have been observed, suggesting the implication of this system in their physiopathology. For instance, rare heterozygous variants in the diacyl glycerol lipase A (*DAGLA*) gene encoding diacylglycerol lipase alpha were significantly associated with seizures and neurodevelopmental disorders, including autism and abnormalities of brain morphology [25].

Understanding the molecular alterations of the ECS in these disorders would be helpful in identifying biomarkers and new therapeutic targets, improving the pharmacological treatments of these psychiatric disorders. Hence, this review summarizes the main findings on the molecular alterations in ECS components at genomic, proteomic, central, and peripheral levels and its detection methods in psychiatric and neurodevelopmental disorders, such as schizophrenia [26], anxiety disorders (AD), depressive disorders (DD) [27], substance use disorders (SUD) [11], ED [28], attention deficit hyperactivity disorder (ADHD), and autism spectrum disorders (ASD) [29] to examine the implication of these changes on the expected performance of the CNS. In each section, polymorphism gene and protein changes will be included, discussing these alterations’ involvement in the development of psychiatric diseases. For this purpose, the endocannabinoid component changes found in animal and human studies of psychiatric diseases will be exposed. 

## 2. A Brief Overview of the ECS

ECS has an essential role in participating in multiple intracellular signaling pathways [30,31]. This system is comprised of endogenous ligands, degrading and synthesizing enzymes, and cannabinoid receptors present in the central and peripheral nervous system [32,33] (Figure 1). ECS is involved in the regulation of physiological functions by the modulation of distinct neurotransmitter systems [32,34]. 

ECBs ligands are lipid messengers acting as agonists of CB1r and CB2r. The two main eCBs are AEA [33] and 2-AG [35]. AEA acts as a partial agonist while 2-AG is a full agonist of both CB1r and CB2r [30,31,32,36,37,38].

AEA is synthesized by the N-acylphosphatidylethanolamine specific phospholipase D (NAPE-PLD) that hydrolyzes AEA [38,39]. AEA is quickly uptaken by the eCBs membrane transporter (EMT) [40]. AEA is degraded by fatty acid amide hydrolase (FAAH), present in the brain at postsynaptic location [41,42]. FAAH is metabolizes AEA into arachidonic acid and ethanolamine [43]. 2-AG is an intermediate metabolite for lipid synthesis and participates in CB1r-dependent retrograde signaling [43], synthesized by the activation of Gq protein-coupled receptors (GPCRs) [38]. The degradation of diacylglycerol (DGLs) precursors by DGL lipases (DAGL-α and DAGL-β) drives 2-AG synthesis [44,45]. The serine-hydrolase enzyme monoacylglycerol lipase (MAGL) catalyzes 2-AG into arachidonic acid and glycerol [41,46].

CB1r is the most abundant G protein-coupled receptor in the brain [47] mediating the physiological actions of eCBs in the CNS [48]. It is widely and heterogeneously expressed in the brain having a crucial role in the regulation of important brain functions [49,50,51]. On the other hand, CB2r was first considered as a peripheral cannabinoid receptor since is highly expressed in the rat spleen [52] and leukocyte subpopulation in humans [53], regulating the immune system [54]. The presence of CB2r in the CNS was only detected under pathological conditions such as in senile plaques in Alzheimer’s disease [55], activated microglial cells/macrophages in multiple sclerosis, spinal cord in amyotrophic lateral sclerosis [56] and the vicinity of tumors [57]. However, CB2r expression in neurons of the brainstem of mice, rats, and ferrets under normal conditions was first described by Van Sickle and colleagues [58]. This finding significantly increased the interest of CB2r in the regulation of brain function [16,59,60,61,62]. Interestingly, CB2r was detected not only in microglia [63] but also in neurons [62,64,65] and astrocytes [66]. Ryberg et al. showed that AEA and CBD act on the orphan receptor GPR55, postulating it as a novel cannabinoid receptor [67]. However, there are several differences between the pharmacology of GPR55 and CB1r and CB2r, so it is still necessary to determine their role in certain disorders since the literature is scarce [68].

Apart from the classical membrane cannabinoid receptors, it is relevant to point out that eCBs (i.e., AEA and 2-AG) could also activate nuclear receptors, particularly α and γ subtypes of peroxisome proliferator-activated receptors (PPARs) [69]. Several mechanisms involved in the interaction between eCBs and PPARs have been proposed. These include direct binding, conversion to metabolites active on these receptors, or the activation of intracellular signaling cascades that ultimately modulate the activity of PPARs [70]. Notably, the endocannabinoid-mediated activation of these nuclear receptors, together with the modulation of canonical cannabinoid receptors, has been involved in the well-known effects of cannabinoids such as analgesia and anti-inflammation neuroprotection or emotional regulation, among other functions [71,72]. Finally, some studies have suggested that other nuclear hormone (non-steroid and steroid) receptors could be implicated in the endocrine actions of cannabinoids, although it seems that hormonal effects are related to the regulation of the endocrine system’s cannabinoid receptors [71].

## 3. Methods to Identify Alterations in the ECS

Identification of changes in the ECS components in animal models and samples from patients affected by the disease is a fundamental tool for clarifying the role of this system in both the pathophysiology and treatment of the disease. Moreover, these alterations may be potential biomarkers for improving diagnosis, prognosis, and clinical outcomes. The following section summarizes the main methods and techniques used for determining modifications of the different critical elements of the ECS: eCBs ligands, enzymes of synthesis and metabolization, and cannabinoid receptors (Figure 1).

### 3.1. Alterations in Endocannabinoid Ligands

Changes in eCBs, such as AEA and 2-AG, are commonly measured by liquid chromatography-tandem mass spectrometry (LC-MS/MS) and its variants [73,74,75], such as ultra-high-performance liquid chromatography coupled to a tandem mass spectrometry detection (UHPLC-MS/MS) [76,77], high-performance liquid chromatography coupled to tandem mass spectrometry detection (HPLC-MS/MS) [78], ultra-performance liquid chromatography coupled to mass spectrometer (UPLC-MS/MS) [79], the ultra-high-performance liquid chromatography-quadrupole time-of-flight mass spectrometry (UPLC-TOF/MS) [80], and ultra-fast liquid chromatography coupled with tandem mass spectrometry detection (UFLC-MS/MS) [81]. These techniques can be applied to different samples (plasma, brain regions, skeletal muscle, adipose tissues, etc.).

For example, using this technique, AEA, palmitoyl-ethanolamide (PEA), oleoyl-ethanolamide (OEA), 2-AG and its isomer 1-AG [78], and N-acyl-ethanolamines (NAEs) and 2-monoacylglycerols (2-MAGs), congeners of AEA and 2-AG [82,83], have been determined in plasma. Similarly, eCBs and their derivatives have been measured in rat liver tissue [84] and AEA metabolite formation in kidney tissue samples [85] by HPLC-MS/MS. Another example is the measurement of AEA, PEA, 2-AG and OEA levels in human hair by UFLC-MS/MS [81].

### 3.2. Alterations in the Enzymes of Synthesis and Metabolization

#### 3.2.1. Genomic Alterations

Several studies have explored genetic alterations in genes encoding synthesis and metabolization enzymes, with the *FAAH* gene the most studied. Several single nucleotide polymorphisms (SNPs) of each enzyme have been identified, some of which have been linked to changes in enzyme function and certain diseases.

SNPs are the most common genetic variations in humans that represent a difference in a single nucleotide that can be replaced, removed, or added to a polynucleotide sequence [86,87]. The SNP may occur in the gene’s coding sequence, in non-coding regions of the gene, or in the intergenic regions between genes. Methods for determining SNPs are diverse and include array-based hybridization (SNP array), polymerase chain reaction (PCR), and sequencing, which are used to evaluate known SNPs [88]. SNP arrays rely on allele-specific oligonucleotide probes. In this case, the genotypes are inferred based on the interpretation of the hybridization signal.

Next-generation or massively parallel sequencing (NGS), as whole-genome sequencing (WGS), whole-exome sequencing (WES), and genome-wide association studies (GWAS) has identified a complete spectrum of genomic alterations, making it possible to carry out polygenetic risk score analysis to understand multiple disorders [89]. Examples of additional genomic alterations are the variability of short sequence repeats (STRs), haplotypes, deletions or insertions of (a) single nucleotide (s), copy number variations, and cytogenetic rearrangements (translocations, duplications, deletions or inversions) [90].

For the enzymes of synthesis, DAGLα and NAPE-PLD, GWAS studies revealed different SNPs. In the case of the DAGLα, approximately 33 SNPs have been identified, for example, rs102275, rs174547, rs198442, and rs199764983. The last one involves the substitution of G by C in the 3049 position (p. Asp1017His). This variation produces a missense variant studied in ADHD [91].

Some of the polymorphisms detected for the NAPE-PLD gene include rs1047998, rs148266530, rs180725393, rs56196003, and rs62482405. Additional SNPs, such as rs17605251 and rs11487077 have been associated, for example, with severe obesity (OB) [92].

SNPs of FAAH gene have been identified by GWAS including rs1571138, rs324418, rs324420, rs4507958, rs324420, rs2275426, rs55867821, rs116435220, rs11576941, rs45512099, rs190112169, and rs17102247. The main SNP for the FAAH gene studied is rs324420, in which a conserved (C) proline (AA129) is substituted with an (A) threonine, which makes FAAH more susceptible to proteolytic degradation [93]. Consequently, A allele carriers reduce FAAH protein levels and activity and increase AEA concentrations [93,94]. These SNPs have been linked to increased susceptibility to drug abuse and dependence [95], bipolar and major depression [96], and stress reactivity [97].

Interestingly, the consequences of *FAAH C385A* appear to depend on its interaction with other SNPs such as rs12075550, which is in a non-coding region near the *FAAH* gene, affecting activator protein binding and expression of this gene [98,99]. The rs12075550 has been linked with specific drug consumption phenotypes [100].

To date, several SNPs of *MGLL* have been identified such as rs9289300, rs9289301, rs9755467, rs113761591, rs113761591, rs2955083, rs2955083, and rs11709060. For example, the rs507961 in *MGLL* was significantly associated with alcohol use disorders (AUD) in adolescents [101]. Similarly, the polymorphism rs604300 has been related to epigenetic modulation of *MGLL* expression. This study reported a higher correlation between childhood adversity, cannabis-dependent symptoms, and this SNP [102].

#### 3.2.2. Epigenetic Alterations

Epigenetic alterations in genes encoding enzymes of synthesis and metabolization of the ECS can also be measured thanks to advances in experimental and computational approaches, such as the accelerating development of arrays and sequencing technologies.

Epigenetics is the biological process that modifies gene expression without affecting the desoxyribonucleic nucleotide acid (DNA) sequence. Notably, epigenetic modulation of SE in biological tissues such asendocannabinoids, phytocannabinoids, and cannabinoid receptor agonists and antagonists induce epigenetic changes with the possibility of transgenerational epigenetic inheritance in offspring. Epigenetic modifiers of the SE could also be a promising tool for treating pathological conditions involving alterations of the SE system [103]. Thus, studying epigenetic alterations is essential to understanding the gap between genotype and phenotype [104].

DNA methylation can induce epigenetic modifications, occurring when a methyl group is added to the fifth carbon of cytosine located within cytosine-guanine dinucleotide (CpG) islets in the gene promoter. This modification changes the chromatin structure from an opened (transcriptionally active) to a closed (transcriptionally inactive) state. Thus, DNA methylation was associated with transcriptional repression [105].

In addition, epigenetic alterations in histones are crucial epigenetic marks [106]. They significantly impact DNA replication and gene expression due to their essential role in the transition between active and inactive chromatin states and controlling epigenetic silencing gene regulation.

Using different techniques, it is possible to measure DNA methylation and histone modifications in genes encoding NAPE-PLD, DAGLα, MAGL, and FAAH. For example, DNA methylation at the *FAAH* gene was detected in peripheral blood mononuclear cells (PBMCs) from subjects with late-onset Alzheimer’s disease, associated with the increased protein levels and activity of FAAH [107]. DNA methylation in CpG sites in the FAAH gene was identified in human saliva after alcohol intake and exercise [108].

For example, histone acetylation at the FAAH promoter was reported in the HYP of rats exposed to binge-eating episodes, down-regulating selectively *FAAH* gene expression in this brain area [109].

#### 3.2.3. Gene Expression Alterations

It is possible to measure alterations at the gene expression level of each enzyme for synthesis and metabolization of the ECS, primarily by real-time q-PCR (quantitative PCR) [110]. Q-PCR has allowed for the evaluation of the ECS in a wide variety of samples and pathologies.

For example, FAAH and MAGL gene expression were measured by q-PCR in plasma and brain regions (prefrontal cortex, PFC; and hippocampus, HIPP) of a rat genetic model of depression [111]. Similarly, gene expression of NAPE-PLD was also assessed using q-PCR [112]. This technique also allows for the measurement of FAAH and MAGL in several brain regions [66,113] of mice, the spinal cord, and skin [114].

Another technique that has gained prominence in recent years is RNAscope. This is a spatial RNA in situ hybridization with a unique probe design strategy that allows for simultaneous signal amplification and background suppression to achieve single-molecule visualization while preserving tissue morphology. This approach provides valuable spatial and temporal information about gene expression in a specific anatomical structure or cell type [115]. This new technique has made it possible to detect MAGL mRNA in the medial septum-diagonal band of Broca area (MSDB) GABA neurons [116].

#### 3.2.4. Protein Level Alterations

Another technique used to study specific protein levels of NAPE, DAGL, FAAH, and MAGL is western blot (WB), a widely employed immunoassay used to quantify the signal emitted by the protein band interest [117]. WB specificity is achieved by using an antibody that recognizes and binds to a unique epitope of the protein of interest. It helps study the presence or absence of proteins (qualitative studies), relative abundance, relative mass, presence of post-translational modifications (PTMs), protein-protein interactions, and less efficiently for quantitative studies [118].

This technique is of great interest for basic research, making it possible to analyze various samples, from cell cultures to different tissues. In the case of the ECS, protein levels of NAPE, DAGL, FAAH and MAGL were measured by WB in the PBMCs of patients with the first episode of psychosis [119].

Immunohistochemical techniques have also been used to study protein alterations of enzymes within the ECS. This technique is based on specific binding between an antibody and an antigen to detect and localize antigens in cells and tissues, most often seen and examined under a light microscope [120]. DAGL immunohistochemistry allowed for the characterization of the enzyme distribution in different human tissues [78]. An example of immunohistochemical studies that have focused on NAPE is evaluating its expression in endometrial carcinoma [121] or studying how it varies according to cannabis consumption [122]. Immunohistochemical studies were carried out to locate FAAH and its distribution in the nucleus accumbens (NAcc) of vervet monkeys [123], the porcine claustrum [124], and human testicular tissue [125], among others.

Finally, MAGL can also be examined using this technique, as evidenced by studies determining the enzyme in colon carcinogenesis or endometrial cancer [126].

#### 3.2.5. Alterations in Protein Activity

In addition to measuring gene and protein levels, it is possible to assess the degree of activity of the different enzymes of synthesis and metabolization.

In the case of DAGL, responsible for the formation of 2-AG, a highly sensitive radiometric assay using 1-oleoyl[1-14C]-2-AG as a substrate can measure its activity. This is based on the use of methods allowing for lipid extraction, fractionation by thin-layer chromatography (TLC) and quantification of radiolabeled [14C]-oleic acid by scintillation counting [127].

Similarly, the activity of NAPE-PLD, responsible for the synthesis of AEA, can be measured by assays based on radioactive substrates and product separation by TLC [128].

To measure MAGL activity, assays can be performed using deuterium-labeled 2AG (d8-2AG) such as the MAGL substrate and measure deuterium-labeled AA (d8-AA) as the MAGL product in biological samples [129]. Furthermore, the enzyme activity can also be measured using fluorometric methods, as already discussed. This case is based on glycerol production from the hydrolysis of 2-AG using membrane preparations overexpressing MAGL from HEK293T cells [130].

#### 3.2.6. Functional Alterations by Neuroimaging Techniques

Another technique to study enzymes is positron emission tomography (PET). Similar to [^11^C]8, new ligands have been developed and investigated to image FAAH [131]. In addition, a wide variety of tracers of different types and acting under other conditions are available for FAAH and MAGL, allowing multiple studies to be performed using this technique [132].

### 3.3. Alterations in Cannabinoid Receptors

#### 3.3.1. Genomic Alterations

Most of the data for genomic alterations in the ECS are on the CB1r. Several polymorphisms for *CNR1* have been described such as rs10498963, rs884647, rs75205693, rs6933130, rs147997421, and rs202070651, among more than 10,000. A significant number of studies have examined the role of different SNPs for the *CNR1* in multiple disorders. For example, the study of rs2023239 in regular cigarette smokers revealed that the C allele variant experienced a lower nicotine reinforcement, suggesting the role of this SNP in nicotine dependence [133]. Another interesting example is the study analyzing SNPs for *CNR1* and personality traits, which associated the rs806372 and rs2180619 with extraversion [134].

In the case of *CNR2*, GWAS studies revealed different SNPs affecting this gene such as rs3003334, rs2229586, rs75459873, rs67934705, and rs11585386. The most studied SNP affecting the *CNR2* is the rs35761398 which substitutes glutamine (Q) 63 with arginine (R) and reduces the function of the gene product as it reduces the efficacy of 2-AG in cells expressing the polymorphic receptor (43). Additional SNPs are rs144279977 and rs201210941, which involve mutations that do not affect the amino acid produced. A different SNP is rs61996280, consisting of a G substitution for A in position 197 (p. Arg66Gln), resulting in a meaningless codon. This SNP was associated with comorbidity of schizophrenia and cannabis dependence [135], drug abuse [136,137], multiple sclerosis [138] and depression [139].

#### 3.3.2. Epigenetic Alterations

Identification of epigenetic alterations of cannabinoid receptors has been paid particular attention, with the gene encoding the CB1r being the most studied [140]. Using microarrays or NGS techniques, it is possible to learn how environmental factors, lifestyles, or diseases induce epigenetic alterations in cannabinoid receptors. In this respect, cumulative evidence indicated that different drugs (alcohol, tobacco, etc.), diet or exposure to stressful situations, among others, induced epigenetic alterations of cannabinoid receptors.

One of the main epigenetic marks that can be measured is the methylation rate of the *CNR1* promoter, related to an increase or decrease in *CNR1* gene expression. For example, in human PBMCs of THC smokers or cigarette smokers, promoter methylation is negatively correlated with CB1r expression level [141]. Similarly, the *CNR1* promoter appears to be mainly repressed by CpG methylation in hippocampal cells [142]. Epigenetic alterations in histones, including acetylation and methylation were also detected after ethanol exposure in different brain regions of the rodents [143,144,145].

In the case of CB2r, its gene has two known isoforms, *CNR2A* and *CNR2B*. The first one has a promoter that contains CpG islands and several CCAAT boxes with a binding site for transcription factors related to stress response [146]. Consequently, the epigenetic regulation of *CNR2* loci by, for example, DNA methylation or histone alterations, might play a crucial role in regulating CB2r gene expression, which deserves further exploration.

#### 3.3.3. Gene Expression Alterations

Most studies have focused on the determination of receptor gene expression. CB1r and CB2r gene expressions can be measured by q-PCR in different types of samples [147,148]. For example, CB1r expression was measured in the dorsolateral frontal cortex (DLPFC) of patients with various psychiatric diseases such as schizophrenia [142,149]. Another example is the quantification of CB2r in the NAcc [150], striatum, substantia nigra (SN), and putamen (PT) in mice [66,113].

In situ hybridization that uses target-specific riboprobes is another technique for quantifying cannabinoid receptors. This technique has been used to quantify the CB1r in both rat and mouse brain tissue samples and different brain regions such as the PFC, secondary motor cortex, dorsolateral striatum, and HIPP areas [151]. It is also used to measure CB2r in renal samples and cell cultures [152].

#### 3.3.4. Protein Level Alterations

WB widely measures CB1r and CB2r protein alterations, for example, to determine the levels of CB1r protein in the spinal cord and dorsolateral periaqueductal grey matter of rodents [153].

In humans, WB has also been used to determine the protein levels of different elements of cannabinoid receptors using central (brain) and peripheral samples (blood samples). For example, protein expression of the CB2r and the non-cannabinoid receptor G protein-coupled receptor 55 (GPR55r) has been measured in *post-mortem* brain tissue of suicide completers [64]. Protein levels of CB1r were examined in *post-mortem* caudate nucleus, Hipp and cerebellum of alcoholic subjects using WB [154]. Similarly, protein levels of CB2r, CB1r and the metabolizing enzymes FAAH and MAGL were measured by WBs in the motor cortex of motor neuron disease patients [155]. Another example is the evaluation of CB1r and CB2r signaling in PBMCs from peripheral blood samples obtained from patients with multiple sclerosis by WB for Erk1/2 [156].

Furthermore, CB1r and CB2r can also be determined by immunohistochemistry. For example, concerning the CB1r, immunohistological studies has allowed for the identification of its ultrastructural localization in the PFC and amygdala (AMY) of mouse brain tissue [157] or evaluating how this receptor is affected when subjected to external factors, such as a fatty diet [158].

The design of new antibodies with higher sensitivity also allows for the measurement of CB2r by immunohistochemistry in different samples, including the brain. For example, immunohistochemistry has identified its role in developing osteoporosis [159]. It has also been used in human samples to clarify how it affects the attenuation of nucleus pulposus degeneration [160] and in vascular and cardiac tissue [161].

#### 3.3.5. Alterations in Protein Activity

As far as studies related to the determination of their activity are concerned, to the best of our knowledge, they have mainly focused on the CB1r. To assess CB1r-dependent Gi/o protein activity, functional [35S] GTPgammaS or guanosine 5′-O-(γ-thio) triphosphate (GTPγS) autoradiography may be used [162], a suitable method to study the function of GPCRs in tissue sections. Using this technique, it is possible to detect up or down-regulation of CB1r binding sites in specific brain regions, or different tissues, allowing us to learn the functional degree of CB1r activation. For example, a down-regulation of CB1r binding sites was found in the cortical regions of aged rats by [35S]GTPγS binding autoradiography [163].

#### 3.3.6. Functional Alterations by Neuroimaging Techniques

PET studies allow for the evaluation of functional alterations in cannabinoid receptors. The [^11^C]A-836339 PET technique was used to map the CB2r and how it affects neuroinflammatory processes in rats [164]. In addition, PET tracer [^18^F] RoSMA-18-d6 is used to look at functional and structural changes in a mouse model of cerebral ischemia [165]. It has also been used for CB1r studies, for example, radiotracer [^11^C] OMAR, to examine the alterations found in women with cannabis use disorder compared to healthy patients [166].

## 4. Anxiety-Related Disorders

### 4.1. Generalized Anxiety Disorder

Generalized anxiety disorder (GAD) refers to excessive fear and worry about several events or activities affecting daily performance. Usually, the anxiety and worry’s intensity, duration, or frequency is out of proportion compared to the actual impact or likelihood of the anticipated event [167]. GAD is considered one of the most prevalent AD, with an estimated prevalence of 7.3% of the world population [168,169]. In addition, GAD is correlated and predisposed to the development of other psychiatric disorders, such as depression, post-traumatic stress disorder (PTSD), or panic attack (PA). In the last two decades, several studies have demonstrated the presence of cognitive and affective hyperreactivity, fear, and negative emotional features and the use of worry to prevent mood contrasts that are difficult to manage in patients with GAD [170]. These symptoms could be correlated with neurobiological disturbances in different brain regions involved in proper emotional regulation. However, the exact neuronal mechanism implicated in the GAD remains unclear. Identifying biomarkers to allow for the correct diagnosis, treatment, and prognosis is essential to increase our understanding of this disorder, allowing the design of new and more effective therapeutic strategies.

In this line, ECS has emerged as a neuromodulator system strongly involved in different neuropsychiatric disorders. It modulates hypothalamic-pituitary-adrenal (HPA) axis activity, usually hyper-activated in anxiety-related disorders [171]. Nevertheless, the implication of this system in GAD has not been thoroughly studied. This part of the review summarizes the main findings regarding the involvement of ECS in GAD from a translational point of view, including animal models of anxiety and human studies (Figure 2).

#### 4.1.1. Clinical Studies

Some studies have identified polymorphisms in genes encoding for ECS components that could increase or decrease the vulnerability to developing GAD. In a study by Demers and colleagues, there was a correlation between two polymorphisms, *FAAH* (rs324420; C285A) and *CNR1* (rs110402), and the risk of developing AD was studied. Authors showed that, as happens in preclinical studies, the increased activity of the FAAH enzyme decreased AEA concentrations in the basolateral AMY, inducing a loss of the inhibitory tone necessary for reducing anxiety and maintaining fear extinction [172]. Low FAAH activity and high AEA concentrations are associated with the A allele of the C385A polymorphism, which showed higher frequency in individuals with increased anxiety and depression scores [173]. This polymorphism affected fronto-limbic connectivity and emerged at 12 years of age, simultaneously with changes in anxiety-related behavior [6]. The same *FAAH* polymorphism (rs324420), combined with the minor allele combination of rs7209436C, rs110402, and rs242924G of the *CNR1* gene, was associated with an increased risk of developing anxiety. *FAAH* gene polymorphism appears to correlate with higher baseline cortisol levels but not anxiety in this study. In addition, baseline cortisol levels could be negatively associated with anxiety [174] (Table 6).

Considering that rs324420 *FAAH* polymorphism alters AEA levels affecting mood regulation and anxiety-like behaviors, another study evaluated the possible correlation between this polymorphism and self-reported anxiety in healthy adults. There was no correlation between both parameters [175]. The presence of *FAAH* C385A and *CNR2* R63Q polymorphisms in a healthy population was evaluated in another study developed by Lazari et al. (2019). In this study, the R allele of R63Q and A allele of *FAAH* C385A were associated with higher depression and anxiety scores and could contribute to greater sensitivity to childhood trauma [176]. On the other hand, *CNR1* rs7766029 polymorphism was correlated with psychosocial adverse event exposure, increasing the likelihood of developing anxiety or depression [177] (Table 6).

#### 4.1.2. Animal Studies

Stress is the major risk factor for developing mood and AD. Different animal models of acute and chronic stress exposure have been developed to evaluate the ECS’s involvement in induced behavioral alterations. These models attempt to simulate some features of GAD, specifically the increased anxiety, considering changes in components of ECS in brain regions involved in emotional regulation under different environmental conditions. Acute restraint stress reduced AEA contents in the AMY by increasing FAAH enzyme activity [24,178]. Similar reduced levels of AEA were found by exposing rodents to a forced swim test as an acute model of anxiety. The inhibition of FAAH activity in the medial PFC increased stress-coping behaviors, indicating that normal ECS signaling is necessary for a proper physiological response to a stressful situation because of its ability to normalize stress response [179]. FAAH increase is consistent with other studies where stress exposure was accompanied by increased FAAH activity and reduced AEA levels. These alterations were accompanied by changes in dendritic arborization, complexity, and spine density in the basolateral nucleus of the AMY and increased anxiety-like behaviors. Thus, FAAH is required to induce hyperactivity and neuronal remodeling of the AMY.

On the other hand, considering chronic stress exposure, a FAAH-mediated decrease of AEA levels following chronic stress is essential to achieving normal HPA functioning [180]. The loss of this activity could be involved in the dysregulation of the HPA axis after chronic stress. In a combined restrain and forced swim test model, stress increased FAAH gene and protein expression in the HIPP but induced a small and non-significant reduction in AEA levels in the PFC. These results are in line with those obtained by Navarria and colleagues. A dual FAAH and TRPV1 inhibitor administration modulated stress-related behavioral alterations and normalized HPA axis activity [171].

Acute and repetitive stress induce essential changes in cannabinoid ligands in different brain regions. Reduced AEA levels were found after the first and fifth restrain stress exposure in the AMY, a brain region strongly involved in mood and emotional regulation. Interestingly, 2-AG levels increase in the AMY and forebrain, but only after a few days of restraining stress [181,182]. Thus, changes in 2-AG concentrations and CB1r content and activity could contribute to repeated and chronic stress [183,184]. Similar results were found in another study developed by Rademacher et al. Both stressful situations (acute and chronic) reduced AEA and PEA protein levels in the AMY and PFC and increased them in the VS, as observed with OEA levels (only after 10 day-daily exposure). 2-AG levels in the AMY increased after 10 days of stress and decreased in the VS after seven days of daily exposure to restrain stress. These results indicate significant ECS alterations in the AMY and PFC, both regions involved in emotional regulation.

Interestingly, some of these changes are only present long-term and occur days after starting with the stress exposure paradigm. This fact, along with the decreased corticosterone delivery after repeated stress, indicates that the ECS is also involved in stress habituation and HPA axis modulation after chronic exposure [185]. In addition, the ECS components are involved in the termination of the stress response, as demonstrated in a study developed by Hill and colleagues. The authors declared that stress induces mobilization of eCBs content within the medial PFC with the participation of glucocorticoid receptors. CB1r activation in this brain region contributes to the termination of the stress response, indicating that alterations on CB1r or eCBs ligands could be involved in the pathological stress response with an impaired termination process [186].

Chronic psychosocial stress exposure (social defeat) increased anxiety-like behaviors and reduced AEA levels in the dorsal caudate-putamen (Cpu), increasing PEA and OEA levels in the same brain region [187]. Similarly, after repeated stress exposure, AEA content was persistently decreased throughout the corticolimbic circuit (including the frontal cortex, HYP, AMY, and HIPP). In contrast, an increase in 2-AG was found in AMY, associated with reduced corticosterone delivery [188]. Interestingly, 2-AG levels were downregulated in the HIPP and CB1r levels, resulting in memory impairment, usually after chronic stress exposure [189]. Thus, repeated stress exposure increased CB1r levels in the PFC and reduced it in the HIPP and AMY of adult rats. Interestingly, in adolescence, increased levels of CB1r in the AMY and PFC were observed. Adolescence stress exposure induced a sustained downregulation of prefrontal cortical CB1r in adulthood, indicating long-lasting consequences of early life stress and its possible correlation with the development of neuropsychiatric disorders in adulthood [190].

In summary, the results obtained in this section indicate the strong involvement of the ECS in chronic anxiety modulating, at least in part, the HPA axis activity. Studies have shown that reduced AEA levels could be associated with increased anxiety, despite the need to develop further reports evaluating this correlation. Future potential therapies treating GAD could focus on increasing endocannabinoid tone.

### 4.2. Post-Traumatic Stress Disorder (PTSD)

PTSD was traditionally classified as an “anxiety-related disorder” until the Diagnostic and Statistical Manual of Mental Disorders Fifth Edition (DSM-V). In this last edition of DSM, PTSD has been postulated as a “trauma- and stressor-related disorder,” mainly because of the complex clinical presentation of this psychiatric disease including anxiety-related symptoms and depression, impulsivity, and psychosis [167]. The central core of the diagnosis of PTSD is exposure to any traumatic and very stressful situations. Patients then develop intrusion and avoidance symptoms and alterations in arousal and reactivity. Despite the high incidence of traumatic exposures worldwide, only 0.7–1.1% develop PTSD [191]. Neuronal anatomical and circuitry changes have been correlated with PTSD symptomatology. However, the molecular alterations underlying these changes and the implication of the ECS remain unclear.

Several animal models have been developed to simulate some features of this complex psychiatric disorder. Nevertheless, most of them present significant limitations regarding their ability to induce intense and long-lasting alterations. In this section, preclinical and clinical studies are included to clarify the role of the ECS in the physiopathology of PTSD (Figure 2).

#### 4.2.1. Clinical Studies

Some authors have studied the implication of the ECS in PTSD, analyzing different polymorphisms of components of this system in patients with PTSD diagnosis. The polymorphism rs1049353 of the *CNR1* gene was associated with an increased likelihood of developing PTSD in a Caucasian population [192]. However, no associations were found in another study evaluating the same target between this polymorphism and PTSD symptoms in war-exposed patients [193] (Table 6).

Interestingly, neuroimaging studies have indicated increased brain CB1r availability in patients with PTSD compared with controls with a lifetime history of trauma and healthy subjects. These changes were accompanied by reduced AEA plasma concentrations in the PTSD group compared to the other two [194]. The same research group investigated the possible correlation between *CNR1* gene polymorphism rs1049353 and PTSD symptoms in children exposed to physical abuse. The results showed a significant interaction between the rs1049353 minor allele (A) and high levels of childhood physical abuse, with more considerable threat symptoms [195] (Table 6). Thus, the abnormal CB1r-mediated signaling could be implicated in the etiology and susceptibility to PTSD after a traumatic experience.

ECS polymorphisms were also correlated with extinction in PTSD patients. More specifically, the minor alleles of rs2180619 and rs1049353 polymorphisms of the *CNR1* gene were associated with poorer extinction learning in PTSD participants. The minor allele of FAAG rs324420 was also associated with worse extinction. Studies developed in healthy volunteers showed that *FAAH* rs324420 effects are dependent on plasma AEA levels and that the minor allele of this polymorphism, in conjunction with higher AEA concentrations, is associated with better extinction learning [196] (Table 6).

#### 4.2.2. Animal Studies

There are different types of approaches to induce PTSD-like behaviors in animals. Usually, and considering the core of PTSD development, the main characteristic of these models is exposure to a very stressful and traumatic situation. For this purpose, psychosocial, physical, or psychological stressors can be used. Predator exposure has been used as an animal model of PTSD, evaluating changes in *CNR1* gene expression in different brain regions one week after exposure. In amygdaloid and frontal complexes, a reduced gene expression of this target has been observed among increased anxiety and fear-related behaviors [197].

Similarly, CB1r levels in the PFC were positively correlated with freezing behavior in a classical fear conditioning paradigm, suggesting the implication of this cannabinoid receptor in fear and anxiety-like symptoms [198]. In the same study, no changes were observed in CB2r levels. These results indicate the strong involvement of CB1r in fear and anxiety in PTSD, as described previously with pharmacological and genetic modulation of this receptor [199]. More recently, our research group developed a long-lasting animal model of PTSD (five week duration), evaluating CB1r and CB2r gene expression in AMY immediately after model exposure and nine weeks later because of the implication of this brain region in mood and emotion. Interestingly, *CNR2* gene expression was increased at both short- and long-term evaluations. However, *CNR1* gene expression was decreased in the shortterm, but enhanced levels were observed nine weeks afterexposure. These data suggest the differential role of both cannabinoid receptors in PTSD-physiopathology and the time-dependent regulation of both targets, probably correlated with alterations in HPA axis normal functioning [200]. These results are in line with those obtained by Sabban and colleagues. In this study, *CNR1* gene expression increased the locus coeruleus (LC), decreasing the basolateral AMY [147]. Thus, the LC-AMY circuit could be significant in developing PTSD-like symptomatology. Some authors have found increased protein levels of the CB1r in the infralimbic region of the PFC and the AMY [201]. CB1 receptor availability was also evaluated in the cerebellum after PTSD-like model exposure. At this point, three-daily tail shock applications induced differential regulation of the gene and protein expression of endocannabinoid receptors in the cerebellum and brain stem. In both sexes, stress application reduced relative gene expression of CB1 in the cerebellum, increasing phosphorylated CB1 (inactive form) only in females. Thus, the present study demonstrated the involvement of CB1rs of the cerebellum in PTSD-like behaviors, showing significant sex differences that could explain, at least in part, the increased susceptibility of females to stress, anxiety, and PTSD [202]. However, in the mild single prolonged stress animal model of PTSD, the quantification of cannabinoid components (AEA, 2-AG, FAAH and MAGL) showed no differences between the model-exposed and control animals in different brain regions involved in mood and emotional processing [203].

Alterations in AEA levels were also reported. After a foot-shock training, different cannabinoid ligand contents were evaluated, revealing increased AEA levels in the AMY, HIPP and medial PFC in higher foot-shock exposed rodents than in lower foot-shock exposed animals. In addition, post-training infusions of FAAH inhibitor URB597, which selectively increases AEA levels in the same brain regions, enhance memory, suggesting that AEA modulates aversive memory consolidation [204]. Similarly, rodents exposed to a foot shock model combined with social isolation as an animal model of PTSD showed significantly reduced AEA levels in the HIPP. In those exposed to an extinction process, a reduction in the freezing time was observed without improving social interaction. However, a normalization in AEA levels was also observed in these animals, suggesting the implication of AEA in the psychopathology of the disease and recovery [204].

Few studies have used mice lacking CB1r to analyze the implication of this receptor in PTSD-like behaviors. In a study carried out by Fride and colleagues, the absence of this receptor (KO) induced a greater vulnerability to stress and PTSD-like features [205]. Additionally, the pharmacological manipulation of this receptor demonstrated that CB1r mediated signaling is essential to facilitate fear extinction [206].

Thus, the current results indicate that the components of ECS are involved in PTSD-like behaviors. Some differences were found in the type of model used to induce PTSD-like symptoms (acute or chronic), the biological determination (protein or gene), or the brain region used to develop these analyses.

In summary, there is not much information about the implication of the ECS in the physiopathology of PTSD. Nevertheless, the results reported indicate that the ECS is involved in fear acquisition and extinction. The modulation of ECS-mediated signaling could be an effective strategy for developing future pharmacological tools for the treatment of PTSD.

## 5. Depression

Major Depressive Disorder (MDD) is a growing global problem, affecting an estimated 3.8% of the population including 5% of adults and 5.7% of adults over 60 years of age [207]. According to the World Health Organization estimation for 2015, the number of people living with depression worldwide is 322 million, and is a significant contributor to suicide deaths [208]. DSM-V describes MDD characterized by distinct changes in affect, cognition, and neurovegetative functions with episodes lasting at least two weeks. Additionally, five or more symptoms must be present during the same episode, with at least one of the symptoms being either depressed mood or anhedonia [209]. Major depression is distinct from usual mood swings and brief emotional responses to daily problems. It can become a severe health problem, especially when it is recurrent and moderate to high intensity, causing great suffering to the affected person and disrupting their work, school, and family activities [210]. In the worst case, it can lead to suicide, the fourth leading cause of death in the 15–29 age group [211].

Pharmacological treatment of MDD entails relevant limitations such as the delayed onset of antidepressant actions and the appearance of significant side effects. The limited success of drug discovery in the context of depression is ultimately linked to an incomplete understanding of the etiology and underlying neurobiology of this disorder. MDD is a multifactorial disorder that involves multiple interrelated causal mechanisms [212,213] including environment, genetics [214,215], and adverse life events such as early life stress [216,217]. Some genetic influences have been associated with the heritability of MDD, estimated at approximately 38% [218].

Numerous studies have shown that deficits in the ECS system may contribute to developing behavioral, physiological, cognitive, and endocrine symptoms of major depression. However, its role in this psychiatric disease has not been fully elucidated. This section includes the most relevant evidence from animal and human studies, providing vital clues about how the ECS components are disturbed in this psychiatric condition (Figure 3).

### 5.1. Clinical Studies

Besides the preclinical clues supporting the critical role of ECS in depression, there is extensive information from clinical studies. Those evaluating alterations in different ECS components in blood samples and *post-mortem* brain tissue have provided relevant results.

Evidence gathered from preclinical and clinical studies evaluating the effects of rimonabant on emotional behavior has been essential to elucidate the potential therapeutic involvement of CB1r in mood disorders [219]. Several clinical trials aimed to evaluate the anti-obesity properties of rimonabant suggested an association with the appearance of depressive symptoms, suicidal ideation or, at worst, suicide [220]. However, it is essential to point out that some confounders could be implicated, such as the inclusion of obese patients with a previous history of depression, a fact that could increase the possibility of detecting depressive symptomatology, including suicidal ideation or suicide in some patients treated with rimonabant [221,222]. Information from animal studies showed a more complex scenario, emphasizing the importance of aspects such as the dose used or the duration of treatment. For instance, some authors described an anti-depressant profile for rimonabant [223,224], whereas others found a depressogenic effect [225,226]. Interestingly, a recent publication noted the opposite actions on emotional behavior between acute or chronic rimonabant administration [227]. Therefore, it is clear that additional experiments are necessary to clarify how the pharmacological manipulation of CB1r may affect the treatment of affective disorders.

Genetic studies, focused on the role of ECS-related polymorphic gene variants strengthen the role of this system in MDD. These studies suggest a certain degree of heritability for MDD [228]. CB1r/CB2r gene polymorphisms have been associated with behavioral features of depression [229]. Polymorphisms affecting the *CNR1* gene showed increased relevance to MDD in participants with higher exposure to adverse life events. For example, the G allele of the *CNR1* gene polymorphism rs806371 is more frequent in individuals with MDD [230]. Similarly, the G allele of the *CNR1* rs1049353 polymorphism has been associated with resistance to antidepressants [230,231]. Alternatively, the minor C allele of rs2023239 showed a protective influence against MDD [232]. However, studies associating genetic variants and MDD have often found inconsistent data. A meta-analysis found no association between *CNR1* rs1049353 or *CNR1* triple repeat with an increased risk of MDD [233] (Table 6).

Associations of *CNR1* SNPs with susceptibility to MDD and response to treatment have been reported in recent research. This study analyzed 181 Han Chinese with MDD and 80 healthy controls. The association of *CNR1* genetic polymorphisms with MDD susceptibility and treatment response was examined in MDD patients sub-grouped responding to antidepressant therapy compared to healthy controls. The results showed a potential role of the *CNR1* rs806367 polymorphism in susceptibility to treatment-resistant depression (TRD). The *CNR1* rs6454674 SNP was also involved in the vulnerability to MDD in patients with TRD and increased inflammatory activity. The haplotype formed by rs806368 and rs806370 SNPs was not involved in the susceptibility to MDD or in resistance to antidepressant treatment. The haplotype comprising rs806366, rs806367, rs806368, and rs806370 SNPs was associated with the vulnerability to develop MDD and resistance to antidepressant therapy. They hypothesized that *CNR1* genetic polymorphisms are associated with a higher probability of developing MDD and within depressed patients with a higher likelihood of resistance to antidepressant treatment [234] (Table 6).

A significant correlation was found between *CNR2* rs2501432 and MDD. This was first observed by Onaivi et al. [139], who reported a meaningful relationship between *CNR2* polymorphism and depressed patients. Likewise, another recent work reported on the link of the *CNR2* R63Q variation to a greater sensitivity towards childhood trauma and overactivation of the HPA axis [176]. Additionally, the meta-analysis performed by Kong clos et al. revealed that rs2501432 *CNR2* polymorphism might be closely associated with depression [233] (Table 6).

*FAAH* gene functional polymorphisms, particularly rs324420 SNP, whichleads to reduced FAAH activity, have also attracted attention to evaluating their role in depression. A study with 858 subjects was carried out to assess the effect of a functional FAAH polymorphism and exposure to early life trauma on the development of anxious and depressive behaviors. The authors concluded that reduced FAAH activity combined with childhood adversity were significantly associated with anxiety and depression in adulthood [173]. Furthermore, Monteleone et al. [96] suggested that the polymorphism in the *FAAH* gene was associated with bipolar disorder and major depression (Table 6).

The first evidence from *post-mortem* brain tissue samples revealed that CB1r protein expression was decreased in the anterior cingulate cortex (ACC) of patients with major depression [235]. Furthermore, Choi et al. showed that CB1r mRNA levels were higher in the PFC of major depression patients [198]. However, a recent study found a lack of CB1r protein expression differences between depressive subjects and paired control patients. The authors highlighted the crucial CB1r-mediated control of glutamatergic signaling due to its contribution to the pathophysiology of depression [236].

Vinod et al. [237] reported increased concentrations of AEA and 2-AG and enhanced CB1r-mediated G-protein signaling in the PFC of alcoholic suicide victims. In addition, recent investigations carried out by Mato et al. [238] revealed that CB1r functionality was upregulated in the PFC of MDD suicide victims. The authors suggested that this change could be modulated by antidepressant intake. Another *post-mortem* study also suggested a potential role for CB1r in the pathophysiology of depression. This study revealed increased CB1r concentrations and CB1r-mediated G-protein stimulation in the PFC of depressed patients compared to normal individuals [239].

Our group analyzed the possible alterations of CB2r and GPR55r in the DLPFC of 15 controls and 18 suicide victims without a psychiatric clinical history or treatment with anxiolytics or antidepressants. CB2r and GPR55r gene expressions were significantly lower (by 33% and 41%, respectively) in the DLPFC of suicide cases. CB2r protein expression was higher, as were CB2-GPR55 heteroreceptor complexes. The results also revealed the presence of CB2-GPR55 receptor heteromers in both neurons and astrocytes, while microglial cells showed no expression. The results suggested that CB2r and GPR55r may play a relevant role in the neurobiology of suicide [66].

Over the past years, an increasing effort has been made to elucidate the alterations in the ECS components in blood samples of patients with depression, mainly eCBs AEA and 2-AG. In a cohort of 28 women with diagnostic criteria for clinical depression and without medication, plasma 2-AG content was significantly decreased. This decrease was negatively correlated with the duration of the depressive episode [240]. Similarly, basal plasma concentrations of AEA and 2-AG were markedly lower in women with major depression (non-treated), indicating a deficit in peripheral ECB activity. The exposure to a stressful situation induced a significant increase in 2-AG levels without modifying AEA [241]. However, another study described increased plasma concentrations of both AEA and 2-AG in depressed patients, and the elevation of 2-AG was significantly associated with selective serotonin reuptake inhibitor (SSRI) antidepressant therapy [242].

Interestingly, the antidepressant-related effects of physical exercise on eCBs levels have also been analyzed. Intense training in control healthy patients significantly increased AEA serum levels correlated with higher BDNF levels, whereas 2-AG concentrations remained stable [243]. In contrast, moderate exercise in women with MDD produced significant elevations in AEA, but not in 2-AG. However, both eCBs presented significant moderate negative associations between plasma changes and mood states [244]. In addition, a correlation between plasma contents of eCBs and blood pressure was found in depressed women, suggesting for the first time that eCBs could play a role in the regulation of blood pressure and, consequently, in cardiovascular risk factors in women with depression [245].

A recent study analyzed the plasma concentrations of 2-AG and AEA and the *CNR1, FAAH*, and *MGLL* polymorphisms as predictors of depression severity six months after suffering a traumatic injury. They found that higher concentrations of 2-AG after trauma predicted greater depression severity six months later. In addition, they found that carriers of one or more copies of the rare allele (G/G or G/A) of the *CNR1* rs806371 SNP experienced more severe depressive symptoms six months after traumatic injury (Table 6). However, they found no evidence that post-injury circulating concentrations of AEA predicted the risk of developing depression [246].

Finally, it is relevant to point out that the magnitude or direction of changes in the ECS components may depend on the severity of depressive symptoms. Indeed, reduced plasma concentrations of 2-AG and/or AEA and decreased density of CB1r in cerebral gray matter glial cells were detected in patients with MDD, whereas increased plasma concentrations of AEA were found in patients with minor depression [247].

### 5.2. Animal Studies

Chronic unpredictable stress (CUS) is a widely validated and reproducible experimental procedure to induce depressive-like behaviors in rodents, extensively employed to understand the role of the ECS in this disorder. Animal models of DD are crucial to evaluate the involvement of ECS components. Hill et al. showed that male Long-Evans rats exposed to the CUS increased CB1r binding site density in the PFC while decreasing in the HIPP, HYP, and NAcc, and had lower concentrations of AEA in all of these brain regions [248]. Furthermore, gender-dependent effects of CUS were analyzed in Sprague-Dawley rats, finding lower and higher CB1r protein expression in males and females, respectively. In contrast, increased FAAH levels were present in both sexes [249]. In addition, further studies employing the CUS procedure specifically focused on CB1r-mediated signaling revealed a significant loss of functional disturbances in the NAcc [250] and in the lateral habenula (LHb) [251].

Moreover, apart from stress-related animal models, the Flinders Sensitive Line (FSL) is a well-known genetic rat model of depression used to exhaustively analyze disturbances in different components of the ECS in specific brain regions and plasma. FSL rats showed reduced and increased levels of 2-AG and AEA in the HIPP, respectively, while it reduced CB1r, FAAH, and MAGL mRNA levels in the PFC. AEA concentrations were increased in plasma, and 2-AG decreased, interestingly reproducing the effects found in the HIPP [111]. Finally, in Wistar Kyoto (WKY) rats, another genetic animal model of depression, increased FAAH and CB1r binding levels while lower levels of AEA were found in the PFC and HIPP [252].

Despite the limited available evidence, CB2r play an essential role in the neurobiology of depression-related disorders, suggesting the possibility of a very promising biomarker [253]. It is likely that, the first evidence suggesting the role of CB2r in depression was a significant reduction in these receptors in the striatum, midbrain, and HIPP in an animal model of depression [139]. Interestingly, our group further evaluated CB2r involvement in depressive-like behavior regulation by a combined genetic and pharmacological approach. First, CB2xP mice (overexpressing CB2r) were characterized by decreased depressive-like behaviors under basal conditions or after exposure to a CUS procedure. Second, the chronic administration of AM630 (CB2r antagonist) blocked the CUS-induced depressogenic effect in stressed mice, an effect that was associated with an up-regulation of CB2r and brain-derived growth factor (BDNF) in the HIPP [16].

A recent comprehensive review by Rafiei and Kolla [254] proposed that FAAH expression is significantly increased in depressive-like phenotypes, and differences in FAAH expression in depressive phenotypes were primarily localized in the PFC, HIPP, and striatum of the animals. They conclude that FAAH may result in an appropriate target for developing new drugs for MDD.

## 6. Schizophrenia

Schizophrenia is a chronic and disabling psychiatric disorder due to genetic causes and impaired brain development in early life. Symptoms of schizophrenia are classified into positive, negative, and cognitive categories. Positive symptoms encompass behaviors and thoughts that are generally not present, such as recurrent psychosis, which is the loss of contact with reality, consisting of delusions, hallucinations, disorganized speech or grossly disorganized or catatonic behavior. Negative symptoms are part of an amotivational syndrome characterized by social withdrawal, affective flattening, anhedonia and diminished initiative and energy. Among many cognitive dysfunctions, cognitive symptoms are expressed as reduced attention or altered speech [209,255,256,257].

Schizophrenia affects approximately 24 million people worldwide, or one in 300 people (0.32%) [207]. Schizophrenia is commonly associated with significant disturbances in personal, family, social, educational, and occupational areas of life and a higher probability of dying earlier than the general population [258,259]. Recent evidence points out that the onset of this mental disorder is most often during late adolescence and tends to happen more frequently and earlier in men than in women.

Currently available pharmacological treatments for schizophrenia have limited efficacy and are sometimes poorly tolerated, with most patients showing substantial deficits in social, cognitive, and occupational functioning throughout their lifetime [260,261]. These facts motivate research focused on elucidating the precise pathophysiological mechanisms involved. Although extensive research has allowed us to identify specific functional, structural, and neuroanatomical alterations, there is still a long way to better understand what phenomena cause the complex and heterogeneous symptomatology of the disorder [262]. In this sense, there is much evidence of the pivotal role of the ECS in schizophrenia neuropathology, mainly due to the THC-induced psychotic symptoms in cannabis consumers [263]. Thus, a great interest has recently been put forward in identifying specific alterations in ECS components that could be useful to provide new biomarkers for preventive, diagnostic or therapeutic purposes [264] (Figure 4).

### 6.1. Clinical Studies

The involvement of certain *CNR1* polymorphisms in schizophrenia has been investigated. Negative results were associated with a single-base polymorphism of the *CNR1* [265], the polymorphism rs1049353 1359G/A [266,267,268,269], or other *CNR1* polymorphisms such as AL136096 [268], rs6454674 [268], rs806366 [270], rs806368 [269,270], rs806376 [270], rs806379 [269], and rs806380 [269]. Despite that, significant associations of rs1049353, rs7766029, and rs806366 *CNR1* polymorphisms were found [271]. On the other hand, Tsai and colleagues suggested that the (AAT)n triplet repeat in the CNR1 gene promoter region was not involved in the pathogenesis of schizophrenia [272], whereas it was significantly associated with the disorganized subtype of schizophrenia [266]. More recently, a consistent association was established between CNR1 polymorphism rs12720071 and poor cognitive performance in schizophrenic patients. Subjects homozygous for the variant (C/C) presented significantly lower motor speed, verbal fluency, attention/processing speed, and reasoning/problem solving than T allele carriers [273] (Table 6).

Associations between concrete *CNR1* polymorphisms and therapeutic response were discovered. In responsive schizophrenic patients, there was an increased G-allele frequency of the rs1049353 polymorphism, leading authors to propose that the G allele of *CNR1* rs1049353 polymorphism could represent a “psychopharmacogenetic” biomarker [269]. Furthermore, the TT genotype of the *CNR1* rs2023239 polymorphism was linked with better negative and positive symptoms in patients with first-episode psychosis [274]. Likewise, carriers of the CC genotype of the rs7766029 *CNR1* polymorphism presented an improvement in verbal memory and attention, while the rs12720071 AG genotype was associated with better executive functions [275] (Table 6).

Interestingly, rs1535255, rs2023239, and rs6928499 *CNR1* polymorphisms seemed to be associated with a lower risk of developing antipsychotic-induced metabolic syndrome. However, the G allele of the rs1049353 (G1359A) CNR1 polymorphism could be associated with a poorer therapeutic response [276] (Table 6). This information provided important clues for potential pharmacogenetic applications to personalize drug treatments for patients with schizophrenia [277].

*Post-mortem* brain studies have revealed relevant data about CB1r protein and gene expression alterations. CB1r availability significantly increased in the DLPFC [26,27,28], although only in paranoid schizophrenic patients [278]. On the contrary, no differences were detected in CB1r-mediated functional coupling to G-proteins in the PFC of schizophrenic and control patients [149]. In addition, CB1r binding levels were more elevated in the left ACC [279] and superficial layers of the posterior cingulate cortex (PCC) [280], but no changes were present in the superior temporal gyrus (STG) [281]. Nevertheless, decreased CB1r protein levels [282,283] and CB1r gene expression [142,149,282,284] were observed in the PFC of schizophrenic patients. Volk et al. revealed lower CB1r mRNA and protein levels in the PFC and increased CB1r binding [285]. Interestingly, upregulation of CB1r gene expression was found in the DLPFC of patients with schizophrenia who died by suicide [142].

The changes observed in PBMCs (e.g., lymphocytes) may be mirrored, to some extent, the neuropathological hallmarks of the disorder. In the first published study, no changes were detected in CB1r mRNA levels in PBMCs from control and schizophrenia patients [286]. Likewise, a similar result was obtained in peripheral immune cells, although a positive correlation between CB1r expression on monocytes and cognitive impairment was present [287]. However, another study detected an increase of CB1r in PBMCs of patients with schizophrenia [288]. Moreover, increased CB1r mRNA levels in PBMCs [289,290] may be associated with lower DNA methylation of the *CNR1* promoter region [290]. Interestingly, a positive correlation was found between CB1r gene expression and positive and negative syndrome scale (PANSS) total symptom severity, which negatively correlated with cognitive functioning [289].

Apart from PBMCs, another interesting biological sample is the olfactory neuroepithelium (ON), a specialized epithelial tissue that is relatively easy to collect that has emerged as a promising tool. In this regard, Guinart et al. identified an increased expression of the heteromer formed between CB1r and the serotonergic 2A receptor (5HT2Ar) in the ON of schizophrenic patients compared to healthy controls. CB1r-5HT2Ar heteromer expression correlated with worse attentional performance, suggesting that it could be a biomarker related to neurocognitive impairment in these patients [291].

Wong et al. analyzed CB1r binding employing [^11^C]-OMAR (JHU 75528), a novel PET tracer, revealing that CB1r binding was significantly higher in the pons of schizophrenic patients [292]. In addition, CB1r binding measured with [^18^F]-MK-9470 PET showed a significant increase in the NAcc, insula, cingulate cortex, inferior frontal cortex, parietal and media temporal lobe of patients with schizophrenia. Interestingly, there was a negative correlation of CB1r binding with negative symptoms and depression scores, especially in the NAcc [293]. However, another study revealed lower CB1r availability levels ([^11^C]-OMAR PET) in the AMY, caudate, PCC, HIPP, HYP and insula of schizophrenic patients [294]. These discrepancies may be related to distinct confounding factors such as symptom severity, sex, age, PET tracer, statistical analysis method or comorbid nicotine use [295].

The group of Oliver Howes recently provided very relevant neuroimaging results on the role of CB1r in patients with FEP. Significant lower CB1r availability was measured by [^18^F]-FMPEP-d2 or [^11^C]-MePPEP PET in patients with FEP, independently of antipsychotic medication treatment. Interestingly, a more significant reduction in CB1r availability was correlated with greater symptom severity and poorer cognitive functioning [296]. In addition, they revealed that the negative association between glutamate levels in the anterior cingulate cortex (ACC) and CB1r availability in the striatum and HIPP present in healthy volunteers were lost in drug naïve patients with FEP [297]. Moreover, another recent work pointed out that striatal CB1r availability was significantly associated with working memory deficits in non-treated FEP patients suggesting critical therapeutic implications [298].

Although the literature is scarce, there is interesting evidence on the potential role of the CB2r in schizophrenia. Specific *CNR2* polymorphisms leading to a loss of function (R63 allele of rs2501432 (R63Q), the C allele of rs12744386, and the haplotype of the R63-C allele) were significantly increased in patients with schizophrenia. Indeed, CB2r mRNA and protein expression were decreased in the DLPFC of schizophrenic patients [299]. Furthermore, the association between three *CNR2* polymorphisms (rs2501432C/T, rs2229579C/T, rs2501401G/A) and schizophrenia was explored [300]. In this sense, a recent study did not find any association between *CNR2* polymorphism rs2229579 and neurocognitive performance in patients with schizophrenia [273]. Likewise, other *CNR2* polymorphisms (rs6689530 and rs34570472) were not associated with schizophrenia in a Korean population [270]. However, *CNR2* polymorphisms rs35761398 and rs12744386 were recently associated with schizophrenia and cannabis dependence comorbidity [135].

CB2r gene expression was significantly reduced in PBMCs from schizophrenic patients in clinical remission [286], and CB2r protein expression was significantly down-regulated together with reduced levels of eCBs synthesizing enzymes (NAPE-PLD and DAGL) in FEP patients [119]. Interestingly, CB2r gene expression was increased in PBMCs [289], correlating with PANSS and cognitive performance severity [287,289], and in cells of the innate immune system [288].

In patients with acute schizophrenia blood, AEA levels were higher and normalized with clinical remission [38]. A significant increase in AEA levels was detected in the cerebrospinal fluid (CSF) of schizophrenic patients [55]. Likewise, an 8-fold increase in AEA levels was found in the CSF of antipsychotic naïve first-episode paranoid schizophrenic patients. In contrast, there were no changes in patients treated with typical but not atypical antipsychotics. Furthermore, there was a negative correlation between AEA levels and psychotic symptoms in non-medicated acute schizophrenics [56].

Similarly, another recent study indicated that AEA and OEA plasma concentrations were increased in schizophrenia patients recruited at the emergency setting and normalized after discharge. This study suggests these endocannabinoid ligands as potential biomarkers of the stressful situation associated with an acute mental crisis before admission to the emergency department [301].

AEA levels were higher in the CSF of schizophrenic patients smoking cannabis. The increase was more than 10-fold higher in low-frequency compared with high-frequency cannabis users [302]. In twin-pairs discordant for schizophrenia [303] or in schizophrenic patients with a comorbid substance use disorder (SUD) [304], higher AEA serum levels were obtained. However, other studies showed no different serum AEA levels [305], increased 2-AG and decreased AEA in the cerebellum, HIPP, and PFC of schizophrenic patients [306]. It is relevant to note that a recent study revealed reduced AEA and 2-AG plasma levels in ultra-high risk psychosis subjects, showing a weak correlation with symptom domains in these individuals who later developed a psychiatric diagnosis [307].

The association of some *FAAH* or NAPDE-PLD polymorphisms with schizophrenia was studied, but no significant results were obtained [271,308]. However, Monteleone and colleagues suggested that the A allele of the 385C/A (rs324420) *FAAH* polymorphism may predispose schizophrenic patients treated with antipsychotics to obtain a clinically meaningful weight gain [309]. More recently, an association between the variant *FAAH* p.Pro129Thr with substance use was found in individuals with schizophrenia [310].

Similar FAAH and MAGL mRNA levels, whereas higher FAAH activity was found in the PFC of schizophrenic patients compared with controls [149]. Interestingly, decreased FAAH mRNA levels were significantly correlated with clinical remission in patients with schizophrenia [286]. In FEP patients, peripheral FAAH and DAGL expression and short-term verbal memory, NAPE-PLD expression and working memory, and MAGL expression and attention were relevant correlations. Finally, 2-AG metabolizing enzyme ABHD6 mRNA levels were higher in patients with schizophrenia [311]. Following these previous results, it was suggested that ECS components could serve as potential biomarkers or pharmacological targets for FEP [312].

### 6.2. Animal Studies

CB1r activation induces sensorimotor gating deficits measured in the prepulse inhibition (PPI) paradigm [313,314], whereas blockade of CB1r presents opposite actions [315,316,317]. Indeed, CB1r pharmacological blockade has shown therapeutic potential in animal models of schizophrenia developed by the antagonism of N-methyl-D-aspartate (NMDA) receptors (NMDAr) by phencyclidine (PCP) or MK-801 [318]. Interestingly, the administration of CB1r antagonists (e.g., AM251, AVE1625) reversed or attenuated PCP- or MK-801-induced pre-attentional, cognitive, or emotional behavioral traits [319,320,321,322]. Therefore, preclinical studies have pointed out the therapeutic implications of CB1r pharmacological regulation in schizophrenia.

In rats exposed to an animal model of schizophrenia induced by the administration of methyl azoxy methanol (MAM) during gestation, decreased methylation of the cannabinoid receptor-interacting protein (CNR1P1) DNA promoter was observed in the ventral HIPP [323,324]. Interestingly, the overexpression of CNR1P1 in the HIPP induced significant schizophrenia-like cognitive and social interaction impairments, as well as an increase in dopamine (DA) neuron population activity in the ventral tegmental area (VTA) [325].

In the so-called ‘three-hit’ animal model of schizophrenia, CB1r binding and cannabinoid agonist-mediated G-protein activation decreased in cortical, subcortical, and cerebellar brain regions [326]. In the MAM model of schizophrenia, CB1r gene expression was reduced in the PFC and increased in the dorsolateral striatum [151]. In addition, CB1r immunoreactivity was significantly increased in the PL, cingulate cortex and cornus ammonium 3 (CA3) regions of the HIPP in the spontaneously hypertensive rats (SHR) strain [327]. Furthermore, CB1r protein expression was positively correlated with the number of involuntary orofacial movements in rats exposed to an animal model of haloperidol-induced tardive dyskinesia [328].

CB2r has also been linked to schizophrenia [329,330]. CB2r blockade by the administration of AM630 significantly exacerbated the MK-801- or methamphetamine-induced hyperlocomotion and PPI disruption, suggesting a CB2r-mediated mechanism [299]. Genetic deletion of CB2r induced some relevant schizophrenia-like features such as increased sensitivity to motor effects of cocaine, anxiety- and depressive-like behavior, disrupted short- and long-term memory consolidation, and impaired PPI [331]. On the other hand, CB2r activation reversed PPI disruptions of the MK-801-induced animal model of schizophrenia. This effect was explicitly mediated by CB2r since only AM630 but not AM251 abolished PPI improvement [332]. Furthermore, activation of CB2r (JWH133) and blockade (AM630) increased MK-801-induced hyperlocomotion, although this effect was much more evident and pronounced with AM630 [333]. Therefore, these results strongly suggest that CB2r functional regulation is significantly involved in schizophrenia-like behavior.

Animal models of schizophrenia have provided exciting results on eCBs brain level alterations. Recently, 2-AG levels were significantly increased in the PFC of Sprague-Dawley rats exposed to the MAM model of schizophrenia [95]. In this regard, a significant increase in 2-AG levels in the PFC of PCP-treated Lister-Hooded rats was reversed by treatment with THC, which induces a substantial reduction in AEA in the same region [91]. Furthermore, in Sprague-Dawley rats exposed to a bilateral olfactory bulbectomy, considered an animal model of depression and schizophrenia, a significant decrease in AEA and 2-AG levels was found in the VS [334]. In addition, mice with a heterozygous deletion of neuregulin 1 (Nrg 1 HET mice), a well-accepted and characterized animal model of schizophrenia [335] display relevant alterations in eCBs levels [336].

## 7. Autism Spectrum Disorder (ASD)

ASD is a neurodevelopmental disorder characterized by deficits in communication and social interaction and a pattern of restricted interests and repetitive behaviors that might vary in severity [167]. It was estimated that around 1.5% of the general population might belong to the autism spectrum [337]. Along with core symptoms, ASD might present several associated problems, such as irritability, challenging behaviors [338], self-injury [339] and intellectual disability (ID) [340]. Other individuals present higher cognitive abilities frequently burdened by psychiatric comorbidities, such as depression, ADHD, or sleep problems [341]. Medical comorbidities are also highly prevalent among ASD subjects [342].

The etiopathogenesis of ASD still needs to be clarified. Several genetic [343], perinatal [344] and environmental factors [345] seem to be involved. Research has also evidenced an imbalance in some endogenous neurotransmission systems [346], including the serotoninergic [347], GABAergic [348], and ECS [349,350]. Imbalances in the EC neurotransmission system were found in subjects with ASD [350] and animal models of ASD [349] (Figure 5).

### 7.1. Clinical Studies

In a pioneer study carried out twenty years ago, reduced CB1r expression was found in the *post-mortem* brains of autistic patients [351]. However, in a recent study with more patients (*n* = 70) [352], the authors found an increased expression of CB1r and CB2r in the PBMC of ASD patients. Interestingly, CB2r but not CB1r gene expression was upregulated in the PBMC of ASD individuals compared to healthy controls [353]. In contrast, gene expression for NAPE-PLD, one of the enzymes responsible for AEA synthesis, was significantly lower in individuals with ASD.

Zou’s study showed that ASD children and valproate-induced rats exhibited reduced cannabinoid content, increased degradation of enzymes, and upregulation of CB1r and CB2r gene expression [340]. More recently, lower circulating endocannabinoid levels in children with ASD were associated with increased degradation enzymes [352,354,355]. Heterozygous rare variants in DAGL-α, the primary enzyme involved in 2-AG biosynthesis, have been associated with ASD [25].

### 7.2. Animal Studies

In the valproate-induced model of autism, brain FAAH levels are increased [349,356], and inhibition of the enzyme attenuates cognitive and synaptic dysfunction [357,358]. These results suggest that reduced AEA and NAE signaling might underlie some of the symptoms of ASD. Accordingly, CB1r activation is also beneficial in the neuroligin-3R451C mouse model of ASD [359], which impaired tonic eCB signaling [360]. Brain concentrations of AEA and NAE were higher in the HIPP of valproate-exposed rats immediately after social exposure [356], suggesting the role of ECS in the adaptative response in ASD. The decreased AEA tone found in ASD (i.e., resulting from a decrease in AEA synthesis by the NAPE-PLD enzyme) may have caused a compensatory increase in CB2r (and probably in CB1r) and a decrease in plasma pro-inflammatory cytokines, thereby supporting the efficacy of CBD in the treatment for ASD [29].

## 8. Attention Deficit Hyperactivity Disorder (ADHD)

ADHD is characterized by inattention and/or hyperactivity or impulsivity [167]. It is a neurobehavioral disorder with a prevalence between 1.4–12% in children and adolescents and between 2.5–5% in adults [361,362,363,364,365,366,367,368,369].

It is a disorder with a multifactorial etiology resulting in a highly relevant heritability of 70–80% [362,364,369,370]. Environmental factors are also essential for the development of ADHD such as pre-and perinatal factors (maternal tobacco and/or substance use; maternal stress; low birth weight); exposure to toxins (e.g., organophosphates); dietary factors (micro-and trace element deficiencies) and factors related to psychosocial adversity (poverty, family conflict) [362,369,371].

Although the exact nature of the neurochemical deficits underlying ADHD is unknown, there is evidence for hypoactivity of frontostriatal dopaminergic circuits [372] and dysfunctional noradrenergic signaling [373]. However, ADHD cannot be explained only by reductions or increases in catecholaminergic systems. Therefore, there is an increasing demand to investigate the alterations in neuromodulator systems such as the endocannabinoid. This section includes the most relevant evidence from animal and human studies, providing vital clues about how ECS components are disturbed in this psychiatric condition (Figure 6).

### 8.1. Clinical Studies

In a pioneer study carried out by our group, allele frequency for *CNR1* was studied in 107 male alcohol-dependent patients (with and without a history of ADHD) and 92 controls. The results revealed a quantitative relationship between more significant alleles of the *CNR1* gene and the diagnosis of ADHD in the patient group [374]. A quantitative correlation between allele sizes and ADHD symptoms was also found. Subsequently, Lu et al., (2008) [192] analyzed SNPs in the *CNR1* gene in two independent samples from Northern Finland: One sample consisting of 187 family trios (ADHD child and two parents) and a second independent sample with 159 adolescent ADHD cases (*n* = 159) and 151 controls (*n* = 151). A significant association was detected for a SNP haplotype (CG) with ADHD diagnosis. It was also found that males were at higher risk than females for this genotype (Table 6).

Ehlers et al. (2007) [375] studied possible SNPs in the *CNR1* gene related to impulsivity. They sought to determine whether a significant association could be detected between the triplet repeat (TAA)n polymorphisms and five SNPs in or near the *CNR1* receptor gene. Impulsivity was significantly associated with the 6-repeat allele of the triplet repeat polymorphism (AATn/A6) and with four SNPs close to the CNR1 receptor gene: rs1535255, rs2023239, rs1049353, and rs806368. The results of this study support the role of the *CNR1* gene in impulsivity (Table 6).

Another study also investigated the association of two *CNR1* polymorphisms with psychosocial adversity and impulsivity in 323 adolescents (170 girls and 153 boys) aged 15 years [376]. Results indicated that impulsivity increased after exposure to early psychosocial adversity and that this effect depended on the *CNR1* genotype. Specifically, early adversity was associated with increased impulsivity among homozygous carriers of the rs806379 A and rs1049353 T allele (Table 6).

Furthermore, the gene encoding FAAH has also been studied. Ahmadalipour et al. (2020) [377] conducted a study with 110 ADHD patients (80 boys and 30 girls) and 100 healthy participants (55 boys and 45 girls). In these participants, the C/T rs2295633 polymorphism of the FAAH gene was determined, and subjects diagnosed with ADHD were found to have an excess of the C allele compared to the control group. The authors suggested that the rs2295633 SNP of the *FAAH* gene could serve as a possible risk marker for ADHD in children (Table 6). Centonze et al. (2009) [378] studied the possible alteration of AEA in subjects with ADHD by assessing the activity of FAAH and NAPE-PLD. This study included 15 children with ADHD (aged 6–13 years) and 15 healthy children. They found a significant decrease in FAAH activity in the peripheral blood of the ADHD subjects compared to the control participants. NAPE-PLD activity was normal, suggesting that only FAAH metabolism is dysregulated. Therefore, it was inferred that AEA levels would be elevated in ADHD.

Brunkhorst-Kanaan et al. (2021) [379] assessed plasma endocannabinoid concentrations in 12 adults with ADHD compared to 98 healthy controls. They explored the levels of AEA, 2-AG, 1-AG, OEA and PEA. They concluded that ADHD patients do indeed present elevated plasma AEA levels. A recent pilot study confirmed that AEA would be increased in subjects with ADHD. Indeed, these findings have served as a basis for studying the efficacy of Sativex THC and CBD in patients seeking treatment for ADHD. The results revealed that this drug improved most of the symptoms related to impulsivity and inattention [380].

### 8.2. Animal Studies

Adriani et al. (2004) [381] studied the delay intolerance paradigm in a group of rats known as the adolescent spontaneously hypertensive rat (SHR) and compared them to Wistar Kyoto rats (WKY, the strain from which SHR are derived). This study evaluated whether WIN 55212, a CB1r agonist, modulates impulsivity in SHR and WKY. The findings revealed that a subgroup of SHR rats presented a phenotype with impulsivity traits. In the impulsive group, the CB1r density in the PFC was significantly lower than in the non-impulsive subgroup and WKYs. Acute administration of WIN 55212 did not affect the performance of WKYs, whereas it increased levels of self-control in impulsive SHRs. These results support the idea that reduced cortical CB1r density is associated with increased impulsivity [380].

Ito et al. (2020) [382] conducted a study suggesting that manipulating endocannabinoid hydrolyzing enzymes is closely related to the symptomatology of ADHD in rats. The experiment was conducted on 56 male Wistar rats and 128 male SHR. They used ethyl octylphosphonofluoridate (EOPF), a potent inhibitor of endocannabinoid hydrolyzing enzymes (i.e., FAAH and MAGL). EOPF treatment decreased FAAH and MAGL activities, increased eCBs levels in the rat brain, and induced ADHD-like behaviors (in the elevated cross maze test) in Wistar and SHR rats. The EOPF-induced behaviors were reversed by concomitant administration of the cannabinoid receptor inverse agonist SLV-319. These findings suggest that activating the ECS is a plausible mechanism for ADHD-like symptoms.

Pattij et al. (2007) [383] investigated the effects of modulation of the ECS on impulsivity. The results showed that the cannabinoid system could modulate inhibitory control. In the inhibitory control test, SR141716A improved inhibitory control dose-dependent by decreasing the number of premature responses.

On the other hand, Leffa et al. (2019) [384] studied the influence of the cannabinoid system on impulsive behavior using a reward delay aversion paradigm and found results that supported the effect of the ECS in modulating reward delay. They found that while a cannabinoid receptor agonist treatment selectively increased impulsive behavior in SHR, a cannabinoid receptor antagonist decreased it. These results are contradictory to those presented by Pattij et al. (2007) and Adriani et al. (2004) in rats and Cooper et al. (2017) in humans, who linked the administration of cannabinoid agonists to decreased impulsivity.

The group of Beltramo et al. (2000) [385] studied the relationship between the dopaminergic and ECS in ADHD. They investigated the effects of the AEA transporter inhibitor N-(4-hydroxyphenyl)-araquidonamide (AM404) on behavioral responses associated with the activation of DA D2 family receptors in rats. They compared the effects of AM404 in SHR versus WKY rats. They found that administration of a low systemic dose of AM404 normalized motor activity in SHR, with no significant motor effect in the WKY controls. These results suggest that pharmacological inhibition of the AEA transporter may alleviate hyperactivity in SHRs.

Castelli et al. (2011) [386] also investigated the relationship between the endocannabinoid and dopaminergic systems in ADHD. They examined the role of CB1r in CB1r-mediated GABAergic synaptic currents and CB1r-modulated glutamate transmission in the striatum. A mouse model of ADHD obtained by a mutation in the DA transporter (DAT) gene was used. DAT-mutated mice presented a marked hyperactive phenotype, and neurophysiological recordings revealed that the sensitivity of CB1rs controlling GABA-mediated synaptic currents in the striatum was lost entirely. In contrast, CB1rs that modulate glutamate transmission were not affected in this ADHD model. These results suggest a complex disruption of the ECS in ADHD, as mice show a specific deficit of endocannabinoid-mediated control of GABA release in the striatum (selective loss of sensitivity of CB1rs controlling GABA transmission), while glutamate control is not affected. Therefore, the inability of striatal projection neurons to suppress inhibition may be directly related to abnormal action selection, a feature of ADHD [387].

## 9. Eating Disorders (ED)

Feeding is a complex and highly conserved process whose orchestration results from homeostatic and hedonic signals [387]. The homeostatic and hedonic feeding components have been attributed to the hypothalamic and reward systems [388]. While the first can be broadly defined as critical regulators of food intake to ensure optimal energy balance, the second mainly relates to the reinforcing properties of sensory stimuli and reward-associated features of feeding.

Evidence that hypothalamic eCBs modulate eating behavior in animals and humans suggests that alterations in ECS may be involved in ED pathophysiology [389] (Figure 7). Further evidence for the notion comes from the observation that CB1r is highly expressed in the areas related to reward, which implicates the ECS in the processes underlying the motivation to eat [390].

EDs are characterized by unhealthy eating. The main EDs are Anorexia Nervosa (AN), Bulimia Nervosa (BN), and Binge Eating Disorder (BED) [167]. In Western countries, the prevalence of BN and AN among the young is 1–3% and 0.5%, respectively. In adolescents, the prevalence of AN and BN is around 0.8% and approximately 5.8%, respectively [391].

### 9.1. Clinical Studies

Genetic studies on EDs have mainly highlighted the role of genetic variants in the *CNR1* gene encoding the CB1r. CB1rs are particularly abundant in areas of the brain related to reward mechanisms. Food restriction or fasting increases eCBs [390]. This fasting-induced increase in endogenous cannabinoids may boost the motivation to eat and the enjoyment of food during ingestion [392].

In a family study by Siegfried et al. (2004) [393] on genetic variants of the *CNR1* gene, fifty-two families (parents with one or two affected siblings) were genotyped for the (AAT) trinucleotide repeat of the *CNR1* gene. The distribution of alleles transmitted to the patients was not significantly different from the non-transmitted parental alleles. However, the 14-repeat allele was preferentially transmitted in the bingeing/purging AN group but not in the restricting AN group. In contrast, the 13-repeat allele was preferentially transmitted in the restricting AN group but not in the bingeing/purging AN group. These findings suggest that restricting AN and bingeing/purging AN may be associated with different alleles of the *CNR1* gene. However, Siegfried’s results could not be replicated by Müller et al. (2008) [394]. Paolacci et al. (2020) [395] proposed that the involvement of AAT trinucleotide repeats in *CNR1* could explain the non-Mendelian inheritance of AN, although further studies are needed to test the differential effect of the various repetitions on the CB1 receptor (Table 6).

On the other hand, genetic variants in *CNR1*, such as the SNP rs1049353, may play an essential role in AN’s etiology. A study on AN and BN by Monteleone et al. (2009) [396] explored the rs1049353 SPN of the *CNR1* gene (1359 G/A). The sample consisted of 134 AN patients, 180 BN patients and 148 healthy controls of average weight. In this study, AN and BN patients had significantly higher frequencies of the rs1049353 A/G SNP in the *CNR1* gene compared to healthy individuals. The A allele of the rs1049353 SNP at *CNR1* A/G was associated with a moderate risk of developing AN and BN (Table 6).

Gonzalez et al. (2021) [397] also found no association between rs1049353 SNP in the *CNR1* gene and the risk of developing AN in a study of 221 AN patients and 396 controls. However, they found that the rs806369-TT genotype and the rs806368/rs1049353/rs806369 SPN haplotype of *CNR1* were associated with lower weight and lower body mass index, respectively.

In addition to studies on genotype variations associated with EDs, research was conducted on the genetic expression of CB1/2rs on EDs. Frieling et al. (2009) [398] identified possible differences in the CB1r and CB2r gene expressions of ED. They included 20 patients with AN, 23 with BN, and 26 healthy women in the trial and found significantly higher levels of *CB1R* gene expression for both AN and BN than the controls (between AN and BN, there was no difference).

Schroeder et al. (2012) [399] sampled 43 people with EDs (20 patients with AN and 23 with BN, of whom four with AN and five with BN had presented self-harming behaviors-wrist slashing), and 26 health care workers without pathology. CB1r gene expression was determined in peripheral blood samples. Patients with EDs and self-injurious history of wrist-slitting exhibited significantly lower CB1r gene expression than patients without wrist-slitting and health care workers. No significant differences were found between patients without a history of wrist-slitting and healthcare workers. Lower CB1r mRNA levels in patients with EDs and self-injurious behavior could be defined by an elevated tone of the ECS, leading to subsequent down-regulation of CB1r, related to desensitization in pain perception. Additionally, they considered that decreased CB1r mRNA in patients with EDs was associated with a more severe course of the disease.

Other researchers have started from a hypothesis contrary to Schroeder’s (2012). Gérard et al. (2011) [400] suggested that the ECS is hypoactive in AN, leading to a chronic compensatory up-regulation of the CB1r. To this end, they used PET imaging with the CB1-selective radioligand [^18^F] MK-9470 in a sample of 14 patients with AN; 16 with BN; and 19 controls. The overall availability of CB1r was significantly increased in cortical and subcortical brain areas of AN patients compared to the controls. Regionally, CB1r availability increased in the insula in AN and BN patients, and the frontal and inferior temporal cortex only in AN [400]. These results were replicated by Ceccarini et al. (2016) [401], who investigated whether the in vivo availability of brain CB1r in areas of the reward circuit and regions related to homeostatic processes was associated with median body mass index (BMI) in patients with ED. As in the previous study, they used PET with the CB1r-selective radioligand [^18^F] MK-9470. The sample consisted of 26 healthy participants and ACT patients including 14 with AN, 16 with BN, 12 with functional dyspepsia (FD), severe weight loss due to loss of appetite, and 12 with OB. CB1r availability was inversely associated with BMI in brain regions involved in homeostasis, such as the HIP and brainstem areas in patients with EDs and healthy subjects. The findings indicate that variations in the ECS in brain regions essential for regulating energy balance are linked to body weight, independently of the existence of pathology [401]. However, CB1r availability was negatively correlated with BMI throughout the mesolimbic reward system. Therefore, they propose that altered levels of CB1r in reward areas could play a role in hedonic eating behavior under pathological conditions and the body weight found in patients [401].

Another study by Monteleone et al. (2015) [402] measured peripheral AEA and 2-AG concentrations in seven patients with underweight AN and seven with weight regained after eating their favorite and non-favorite foods and compared them with healthy controls. They found that in the controls, plasma 2-AG concentrations decreased after both types of meals (in the hedonic meal, the decrease was more significant) whereas, in patients with underweight AN, 2-AG concentrations did not show specific changes either related to the intake itself or the hedonic value. However, in weight-restored patients, 2-AG concentration showed similar increases with both types of meals.

The recent pilot study by Piccolo et al. (2020) [403] on patients with AN contradicts the previous studies. Piccolo et al. (2019) sampled 15 hospitalized AN patients and nine controls. Blood samples were collected after an 8-h fasting period, and AEA and 2-AG levels were measured. 2-AG concentrations did not differ [403]. AEA concentrations were significantly lower in AN than in the controls. Moreover, after recovery, no significant difference was found for eCB levels, contrary to Monteleone et al. (2005).

Regarding binge eating disorder (BPD), Monteleone et al. (2017) investigated AEA and 2-AG responses to hedonic eating in patients with BPD [404]. Peripheral levels of AEA and 2-AG were measured in seven obese patients with EDs before and after eating their favorite and non-favorite foods. We found that plasma levels of AEA progressively decreased after eating their non-favorite food, as observed in Monteleone et al. (2017) with healthy and AN participants. However, AEA levels increased significantly after eating their favorite food, which did not occur either in the controls or in AN patients [404], who maintained a pattern of decreasing AEA regardless of the hedonic value of the food. Plasma 2-AG levels did not differ significantly between the two test conditions. Changes in peripheral AEA levels were positively correlated with the subjects’ feelings about the urge to eat and pleasure during eating. In contrast, changes in peripheral 2-AG levels were positively correlated with the subjects’ feelings of pleasure while eating and with the amount of food.

In the recent study by Yagin et al. (2020) [405], in 180 overweight/obese women, 75 were diagnosed with BED. The results suggest that BED women have significantly higher AEA and 2-AG than BED women diagnosed with BED than women without EDs. These data confirm those found by Monteleone et al. (2005) [406].

The *FAAH* gene has also been linked to overweight and BED. The *FAAH* cDNA 385 A/A (P129T) polymorphism has been identified in overweight individuals in 2667 subjects [407]. In this study, Sipe et al. observed that the homozygous *FAAH* 385 A/A genotype was significantly associated with overweight and OB. The BMI was significantly higher in the FAAH 385 A/A genotype group than in the *FAAH* 385 A/C heterozygote groups. Subsequently, the *FAAH* gene was explicitly studied in BED [407]. Monteleone et al. (2008) [408] found that the *FAAH* cDNA 385C to A gene polymorphism was only altered in people with OB compared to the controls but was not significant in BED. Similarly, Yagin et al. (2020) found no significant differences in the frequency of the A allele of the *FAAH* gene in 180 overweight women with and without BED [405].

### 9.2. Animal Studies

One of the most robust leanness-inducing knockouts in animal model studies is the KO mouse for *CNR1* (Cnr1-/-mice). Cnr1-/-mice show a very significant reduction in body weight when fed standard chow, are resistant to the obesogenic effects of a high-fat diet and show reduced food intake from both standard and high-fat diets [409]. Furthermore, CB1r availability, as in human studies by Ceccarini et al. (2016) and Gérard et al. (2011), appears to be altered in animal models [400,401]. Casteels et al. (2014) studied CB1r binding in vivo in rats using PET with the same [(18)F]MK-9470 radioligand [410]. They used an activity-based anorexia (ABA) model in Wistar rats. Experiments were performed on 80 Wistar rats (23 males and 57 females). It was found that compared to the controls, ABA rats showed an in vivo increase in CB1r availability in all cortical and subcortical areas, which normalized toward baseline values after weight gain. Furthermore, they found that only females showed a region-specific increase in CB1r availability. The relative binding of [(18)F]MK-9470 increased in ABA females’ HiPP, inferior colliculus, and entorhinal cortex. Again, relative [(18)F]MK-9470 binding values normalized with increasing weight. These results agree with those presented in Ceccarini et al. (2016) and Gérard et al. (2011) [400,401].

Collu et al. (2019) also conducted a study using the ABA model in rats. They measured CB1r density in the brains of female Sprague Dawley rats with ABA, focusing on areas involved in homeostatic and reward-related regulation of eating behavior [411]. They observed that CB1r density was decreased in the hippocampal dentate gyrus and lateral HYP. After recovery, it completely reverted to control levels in the lateral HYP [411].

In BED animal models, continuous access to a high-fat diet seems to decrease CB1r gene expression in the NAcc and the PFC; even after periods of abstinence, it remains reduced in the NAcc [412].

Animal studies have also addressed alterations in eCBs levels. Collu et al. (2019) analyzed eCB levels in a rat ABA model. At the end of induction of the ABA model, 2-AG levels were significantly decreased in all brain areas tested except in the CPu and significantly reduced in recovered ABA animals compared to the control rats [411].

In contrast, brain AEA levels were not different between the ABA and control groups at the end of the ABA induction phase. The recovered ABA group exhibited a significant decrease in AEA concentrations in all regions tested compared to the control. Thus, although the levels of both eCBs were significantly reduced, they appeared to have a different temporal regulation in the brain of ABA rats [411].

In BED, alterations in eCBs may be involved in the pathophysiology of BED. Satta et al. (2018) conducted a study in which brain levels of AEA and 2-AG were assessed after inducing binge eating behavior with margarine in female Sprague-Dawley rats [413]. The results show that compared to the controls, in the high restriction group AEA was significantly decreased in the CPu, AMY, and HIPP. Similar to the high restriction group, AEA decreased in the AMY and HYP in the low restriction group. In contrast, 2-AG was significantly increased in the HIPP in the high- and low-restriction groups. In addition, the low restriction group also had decreased AEA in the PFC and increased AEA in the NAcc [413].

In a recent study, Berland et al. (2022) showed that BED elicited compensatory adaptations requiring the gut-to-brain axis, which, through the vagus nerve, relies on the permissive actions of peripheral eCBs signaling. Selective inhibition of peripheral CB1r resulted in a vagus-dependent increased hypothalamic activity, modified metabolic efficiency, and dampened the mesolimbic DA circuit activity, altogether leading to the suppression of palatable eating [414].

Finally, alterations in FAAH expression were also investigated in animal studies. In a study by Pucci et al. (2019) with 64 female Sprague-Dawley rats (eight per condition), they used an intermittent food restriction model and a stress-induced frustration model to elicit binge-like episodes. They observed selective down-regulation of *FAAH* gene expression in rats exhibiting binge eating behavior in the HYP with a consistent reduction in histone 3 acetylation at lysine 4 of the gene promoter. Therefore, histone 3 acetylation at lysine 4 of the *FAAH* gene promoter could be a biomarker of EDs [109].

## 10. Substance Use Disorders

Substance use disorders (SUDs) are chronic and relapsing mental illnesses characterized by compulsive behavior, drug-seeking, and repeated achievement of episodes of intoxication and withdrawal. Diagnostic criteria following the 5th Edition of the DSM-V [209] include taking substances in a more significant amount than originally intended, a persistent desire to cut down or moderate drug use, a more extended period using the drug or recovering from its effects, intense craving, and the development of tolerance or withdrawal symptoms. Although our understanding of the etiology of SUDs remains limited, it is well-established that the rewarding effects of substances of abuse are mediated by the activation of the dopaminergic mesolimbic reward system that includes the ventral tegmental area (VTA) of the midbrain and its projections to the PFC, NAcc, and the AMY [415].

The ECS participates in natural and drug rewards through interaction with the dopaminergic mesolimbic reward system. Moreover, accumulative evidence indicates that the ECS may modulate the dopaminergic reward mesolimbic system directly or indirectly via stimulation of the CB1r and CB2r [416]. This review aims to determine modifications of the ECS that could represent biomarkers associated with the development of SUDs.

### 10.1. Nicotine Use Disorders

#### 10.1.1. Clinical Studies

##### Neuroimaging Studies

We identified two neuroimaging studies. In the first study, Jansma et al. (2013) [417] explored alterations in the reward processing by using THC to challenge the ECS in a sample of subjects with nicotine use disorders (NUDs, *n* = 10) compared to healthy controls (*n* = 11). A functional magnetic resonance imaging (fMRI) was performed while patients participated in a Monetary Incentive Delay (MID) paradigm. Reward activity in the NAcc and CPu during anticipation and reward feedback was compared after THC and PLB administration. A significant reduction in reward anticipation activity was found in the NAcc in subjects with NUD after THC administration compared to the healthy controls [417]. A more recent study compared male frequent chronic cigarette smokers (*n* = 18) to non-smokers (*n* = 28) using PET and [^18^F] FMPEP-d_2_, a radioligand for CB1r. The results showed that smokers presented a lower distribution volume of the brain CB1r than non-smokers. This reduction was found in all brain regions, and did not correlate with years of smoking, the number of cigarettes smoked per day, or measures of nicotine dependence [418] (Table 1).

##### Genetic Studies

Two studies were included in this review. The first study targeted the *CNR1* SNPs rs806379, rs1535255, and rs2023239 as moderators of nicotine withdrawal-related cognitive disruption in 73 smokers. It was found that the SNP rs806379 moderated the effects of nicotine withdrawal so that smokers homozygous for the major allele exhibited more significant nicotine withdrawal-related cognitive disruption [419]. A more recent study investigated the influence of the *CNR1* SNP rs2023239 on nicotine reinforcement and craving in regular cigarette smokers (*n* = 104). It was found that smokers with the C allele variant (CC + CT genotypes) experienced a lower nicotine reinforcement effect compared to those without the C allele (TT genotype). Still, there were no differences between genotype groups regarding cue-elicited craving. Thus, the SNP rs2023239 *CNR1* may play a more significant role in nicotine reinforcement other than cue reactivity [133] (Table 6).

#### 10.1.2. Animal Studies

Several reviews have pointed out that the ECS plays a relevant role in NUDs [419,420]. In animal studies, the involvement of the ECS in the rewarding effects of nicotine has generally been evaluated by using the conditioned place preference (CPP) and intravenous self-administration paradigms. CB1r is particularly important as it has been involved in the rewarding properties of nicotine, the cognitive impairments associated with nicotine withdrawal, and relapse to nicotine-seeking behavior. In contrast, CB2r in NUDs is, to date, still controversial. Regarding AEA and 2-AG, studies have provided conflicting results about their reinforcing properties for nicotine and nicotine withdrawal. However, AEA and 2-AG could have opposite roles in relapse to nicotine-seeking behavior [421].

The role of CB1r in NUDs was carried out by exploring the responses induced by acute and repeated nicotine administration in wild-type (WT) and CB1r knockout (KO mice). Nicotine produced a significant rewarding effect in WT mice, but the response was absent in CB1r KO mice. The mecamylamine-induced abstinence model was also used in chronic nicotine-treated mice to precipitate somatic signs of nicotine withdrawal. Still, there were no significant differences in the severity of nicotine withdrawal between WT and CB1r KO mice [422]. A CPP paradigm study evaluated the role of CB1r and the FAAH in nicotine reward by using CB1r KO, *FAAH KO*, and WT mice. Disruption of CB1r blocked nicotine reward, but, in contrast, genetic *FAAH* deletion enhanced the expression of nicotine CPP. As in the previous study, the expression of spontaneous nicotine withdrawal was unaffected in CB1r KO mice. However, *FAAH KO* mice displayed increased conditioned place aversion in a mecamylamine-precipitated model of nicotine withdrawal [423].

A recent study found that inhibition of MAGL attenuates nicotine CPP in mice through a non-CB1r-mediated mechanism [424]. In this study, genetic (by comparing MAGL KO vs. WT mice) and pharmacologic (by using the MAG inhibitor JZL184) approaches were used to evaluate the role of 2-AG in nicotine reward. It was found that MAGL KO mice failed to develop a nicotine CPP compared to WT mice. In addition, administration of JZL184 dose-dependently blocked the development of nicotine reward in the CPP test. Repeated MAGL inhibition did not induce a reduction in CB1r levels or function. Instead, MAGL inhibition caused a concomitant decrease in arachidonic acid levels in various brain regions of interest, suggesting an arachidonic acid cascade-dependent mechanism.

The role of CB2r in the acquisition of nicotine dependence and the processes of abstinence and relapse was also explored in several studies. In a study published by our group, different experimental paradigms were employed to assess nicotine reinforcement in the CPP paradigm, motivation (intravenous self-administration), and withdrawal (mecamylamine-precipitated withdrawal syndrome after chronic nicotine exposure). The role of CB2r was evaluated by comparing CB2r KO mice (genetic approach) and WT mice treated with the CB2r antagonist AM630 (pharmacological approach). CB2r KO mice did not show nicotine-induced place conditioning and hardly self-administered nicotine compared to WT mice. In addition, AM630 blocked nicotine-induced CPP and reduced nicotine self-administration in WT mice. Somatic signs of nicotine withdrawal significantly increased in WT but were absent in CB2r KO mice [425]. These results were partially replicated in another study in which nicotine-induced CPP was entirely blocked by the selective CB2r antagonist, SR144528, in WT mice and was absent in CB2r KO mice.

In contrast to the previous study, WT and CB2r KO nicotine-dependent mice showed almost identical precipitated withdrawal responses, and deletion of CB2r did not alter the acute somatic effects of nicotine [426]. These results suggest that CB2rs are required for nicotine-induced CPP, while its role in nicotine withdrawal or acute effects of nicotine is still controversial. Finally, a recent study [427] (Table 1) reported the absence of nicotine-induced CPP in dopamine transporter gene (DAT) and *CNR2*, *DAT-CNR2* KO mice, similar to previous findings.


ijms-23-04764-t001_Table 1Table 1Summary of the main findings regarding the alterations of the ECS components in animals and human studies in Nicotine Use Disorder.Nicotine Use DisorderAuthorsType of SampleType of EvaluationOutcomes[417]HumansNeuroimaging (fMRI)↓ reward anticipation activity in the NAcc after THC administration in NUDs[418]HumansNeuroimaging (PET)↓ CB1r in all brain areas in NUDs[422]AnimalsCB1r KO vs WT miceNicotine rewarding effects in WT mice but not in CB1r KO miceNo significant differences in the severity of nicotine withdrawal between WT and CB1r KO mice[423]AnimalsCB1r KO vs *FAAH* KO vs WT miceCB1r KO mice blocked nicotine reward*FAAH* KO mice had an enhanced expression of nicotine rewardNicotine withdrawal was unaffected in CB1r KO mice, *FAAH* KO mice displayed increased nicotine withdrawal[424]AnimalsMAGL KO vs WT miceMAGL KO mice failed to develop a nicotine CPP compared to WT mice[425]AnimalsCB2r KO vs WT miceCB2r KO mice did not show nicotine-induced PCC and hardly self-administered nicotine compared to WT miceSomatic signs of nicotine withdrawal ↑ in WT but were absent in CB2r KO mice[426]AnimalsCB2r KO vs WT miceNicotine-induced CPP was absent in CB2r KOWT, and CB2r KO nicotine-dependent mice showed a similar response during nicotine withdrawal[427]Animals*DAT-CNR2* KO vs WT miceCompared to WT, DAT-*CNR2* KO mice showed the absence of nicotine-induced CPP.


### 10.2. Alcohol Use Disorders (AUD)

#### 10.2.1. Clinical Studies

##### Neuroimaging Studies

Several neuroimaging studies have demonstrated the implication of the ECS in AUDs. The most robust evidence comes from studies related to the CB1r. In a pioneer study carried out by Neumeister et al. (2012) [428], PET and the CB1r selective radiotracer [^11^C] OMAR were used to assess CB1r density during early abstinence (four weeks) in men diagnosed with AUD (*n* = 8) compared to healthy controls (*n* = 8). Compared to the controls, men with AUD showed elevated CB1r binding, suggesting CB1r upregulation in a circuit that included the AMY, HIPP, PT, insula, anterior, and posterior cingulate cortices, and orbitofrontal cortex (OFC). No correlations were found between CB1r binding and the age at onset of drinking, the number of drinks, or illness duration [428]. A subsequent study used PET together with [(18)F] FMPEP-d2, a radioligand for CB1r, to assess CB1r in alcohol-dependent subjects (*n* = 18) compared to healthy controls (*n* = 19) in early (3–7 days) and protracted abstinence (2–4 weeks). In contrast to the previous study, CB1r binding, which was negatively correlated with years of alcohol abuse, was 20–30% lower in alcohol-dependent subjects than in healthy controls in all brain regions during the first scan (3–7 days of abstinence). It remained similarly reduced in alcohol-dependent subjects during the second scan (2–4 weeks of abstinence) [418].

A more recent study evaluated the effect of acute and chronic alcohol exposure by using PET to determine the availability of CB1r after a binge-drinking episode, after chronic heavy drinking, and after one month of abstinence. For this purpose, to assess the short-time effects of a binge-drinking episode on CB1r availability, healthy social drinkers (*n* = 20) underwent PET at baseline and after intravenous ethanol administration, and alcohol-dependent subjects (*n* = 26) underwent sequential CB1r PET after chronic heavy drinking and after one month of abstinence and compared to healthy subjects (*n* = 17). Whereas acute alcohol consumption resulted in a global increase in CB1r availability, a global decreased CB1r availability was found in chronic alcohol drinkers compared to the controls. This global decreased CB1r availability remained unaltered after one month of abstinence. These findings were more noticeable in the cerebellum and parieto-occipital cortex following chronic alcohol drinking. After abstinence, they were also extended to other areas such as the VS and the mesotemporal lobe [429]. Altogether, these studies point to the role of CB1r in AUDs. It is possible to find that CB1r increased binding during acute drinking, whereas during chronic drinking and abstinence, CB1r binding is reduced.

##### Genetic Studies

Genetic studies in AUDs were mainly designed considering the SNPs of the *CNR1*. A genetic study that examined the relationship between three SNPs of the *CNR1* rs6454674, rs1049353, and rs806368 and AUD, including alcohol-dependent males (*n* = 298) who were compared to healthy controls (*n* = 155) found an association between the *CNR1* gene and AUDs, so that alcohol-dependent patients with the TGT haplotype (corresponding to SNPs rs6454674, rs1049353, rs806368) were less prone. In contrast, alcohol-dependent males with the GGT haplotype or the TGC haplotype were more prone to develop AUDs [430]. A pioneering study explored the prevalence of the silent polymorphism 1359G/A of the *CNR1* in severely affected Caucasian alcohol-dependent subjects (*n* = 121) compared to the non-alcoholic controls (*n* = 136). The A allele was more frequently observed in alcohol-dependent subjects with severe withdrawal syndrome (42.1%) than in non-alcoholic controls (31.2%). This allelic association resulted from an excess of the homozygous A/A genotype in patients with a history of alcohol delirium, suggesting an increased risk of delirium. These authors concluded that the homozygous genotype *CNR1* 1359A/A confers vulnerability to alcohol withdrawal delirium [431].

Moreover, in another study, male-heavy drinkers (*n* = 80) were genotyped for the *CNR1* rs2023239 polymorphism (CT/CC or TT). Subjective craving for alcohol, subjective arousal, and salivary reactivity were explored after participants were exposed to water and beer in 3-min trials. No substantial evidence was found for alcohol cue-reactivity in the *CNR1* C allele groups. However, as weekly alcohol consumption increased, the *CNR1* C allele group reported more craving for alcohol during alcohol exposure than the T allele group [432]. In addition, several studies investigated the association between SNPs of the *CNR1* and the location and function of the CB1r. Hirvonen et al. (2013) [433] studied the potential influence of the rs2023239 variation in the *CNR1* in alcohol-dependent subjects (*n* = 18). They found that C allele carriers at rs2023239 presented higher CB1r binding than non-carriers [433] (Table 6).

In addition, *CNR1* polymorphisms (rs806368, rs1049353, rs6454674, rs2180619) and *FAAH* polymorphisms (rs324420) were also associated with sleep disturbances in patients with AUD. These sleep disturbances represent risk factors for alcohol relapse. A study explored the influence of these polymorphisms on subjective and objective sleep quality in individuals with AUD (*n* = 497) compared to the controls without this disorder (*n* = 389). It was found that subjective sleep disturbances differed significantly among *CNR1* rs6454674 genotypes in both subjects with AUDs and the controls, but only among the controls, neuroticism personality scores mediated the relationship between genotype and sleep disturbances. On the other hand, objective sleep measures differed significantly by the *CNR1* rs806368 genotype, both at baseline and follow-up, and the *FAAH* rs324420 genotype for recorded sleep duration among individuals with AUD at follow-up [434].

A recent meta-analysis of genetic and genome-wide association studies exploring the correlations between the *CNR1* SNPs, and the risk of alcohol dependence found that three *CNR1* SNPs (rs1535255, rs2023239, and rs806379) were associated with the development of alcohol dependence. The polymorphism rs1535255 presented more significant evidence for its association with AUD. In contrast, the polymorphism rs806379 was explicitly implicated in the Caucasian subgroup [435] (Table 6).

Only one study evaluated the association between the *CNR2* polymorphism Q63R and AUDs in the Japanese population. Alcohol-dependent patients (*n* = 785) and age and gender-matched controls (*n* = 487) were compared. There was a significant association between the *CNR2* Q63R polymorphism and Japanese alcohol-dependents (*p* < 0.01). This polymorphism would be related to reduced CB2r-mediated response and a greater vulnerability for the development of AUD [436] (Table 6).

##### *Post-Mortem* Studies

*Post-mortem* brain studies were used to explore the involvement of the ECS in AUDs. A pilot study examining differences in profiles of the ECS between Cloninger type 1 (*n* = 9) and type 2 (*n* = 8) alcohol-dependent subjects and healthy controls (*n* = 10) found significant differences between these three groups in the EC profiles of the AMY and HIPP. Cloninger type 1 alcohol-dependent subjects showed significantly increased docosahexaenoyl ethanolamide (DHEA) levels in the AMY, and a significant negative correlation between AEA concentrations and mGlu1/5 receptor density in the HIPP compared to Cloninger type 2 alcohol-dependent subjects and controls [437]. Moreover, Erdozain et al. (2015) [438] carried out several *post-mortem* studies on the alcohol-dependent population. In the first study, the state of the CB1r, the enzymes FAAH and MAGL, and the extracellular signal-regulated kinase (ERK) and cyclic-AMP response element-binding protein (CREB) were assessed in the *post-mortem* PFC of alcohol-dependent subjects and controls, who were classified into four different groups: (1) non-suicidal alcohol-dependent subjects; (2) suicidal alcohol-dependent subjects; (3) non-alcohol-dependent suicide victims; and (4) control subjects. Although no statistically significant differences were observed in CB1r mRNA relative expression among the four groups, an increase in CB1r protein expression in the PFC of the suicidal alcohol-dependent group was found. In addition, alcohol-dependent subjects presented lower levels of MAGL activity, a decrease in the active form of ERK, and CREB levels, regardless of the cause of death. In a subsequent study of the same group, the state of CB1r in the *post-mortem* caudate nucleus, HIPP, and cerebellum of alcohol-dependent subjects was evaluated. They found that alcohol-dependent subjects presented hyper-functional CB1r in the caudate nucleus, resulting in a higher maximal effect in suicidal and non-suicidal, alcohol-dependent groups than in non-alcohol-dependent groups. Conversely, in the cerebellum, the non-suicidal alcohol-dependent subjects showed hypofunctional CB1r [154,438] (Table 2).

#### 10.2.2. Animal studies

The CB1r has been associated with the rewarding properties of alcohol, consumption and self-administration alcohol behavior, alcohol tolerance and dependence, and relapse to alcohol consumption. A pioneering study evaluated the role of CB1r in voluntary alcohol consumption and acute alcohol-induced DA release in the NAcc, using CB1r KO and WT mice. CB1r KO mice exhibited reduced voluntary alcohol consumption and completely lacked alcohol-induced DA release in the NAcc compared to WT mice [439]. These results were further confirmed in another study in which male CB1r KO mice displayed decreased ethanol-induced CPP compared to WT mice [440]. Altogether these results suggest that the CB1r plays an essential role in regulating the positive rewarding properties of alcohol.

The role of CB1r in alcohol tolerance and withdrawal was also explored. CB1r KO mice showed decreased ethanol consumption and preference compared to the WT controls. This decreased ethanol self-administration was associated with increased sensitivity to the acute intoxicating effects of ethanol. The severity of alcohol withdrawal-induced convulsions was also increased in CB1r KO mice [441].

Onaivi’s group was the first to report the involvement of CB2r in addiction [139,436]. Using a model of voluntary alcohol consumption (with the two-bottle choice paradigm), they demonstrated that mice experiencing more alcohol preference were characterized by a lower gene expression of *CNR2* in the ventral midbrain. In addition, systemic administration of the CB2r agonist JWH015 enhanced alcohol consumption in mice previously exposed to chronic mild stress [139,436].

Our group extended these results, examining the role of the CB2r on the vulnerability to ethanol consumption using the genetically modified animal’s approach. CB2r KO and WT mice were compared regarding the reinforcing properties of ethanol, the preference and voluntary ethanol consumption, and oral ethanol self-administration. CB2r KO mice presented an increased response to ethanol effects, ethanol-induced CPP, voluntary ethanol intake and preference, acquisition of ethanol self-administration, and motivation to drink ethanol compared to WT mice [442]. These results suggest that genetic deletion of the CB2r could increase the vulnerability of animals to the reinforcing and motivational stimuli of alcohol.

Other components of the ECS have also been studied. Mice lacking the *FAAH* are severely impaired in their ability to degrade AEA and therefore represent a unique animal model to examine the function of AEA on ethanol-drinking behavior. A study found that *FAAH* gene KO mice preferred alcohol and voluntarily consumed more alcohol than their WT littermates. There were no significant differences between FAAH gene KO and WT mice in the severity of ethanol-induced acute withdrawal, conditioned taste aversion to alcohol, CPP, or sensitivity to the hypnotic effect of ethanol. These data suggest that deletion of FAAH genes increases the ethanol preference, decreases the sensitivity to ethanol-induced sedation, and fasters the recovery from ethanol-induced motor incoordination [443].

Interestingly, female FAAH KO mice presented increased ethanol intake and preference and showed no CB1r down-regulation after voluntary ethanol consumption compared to male FAAH KO and male and female WT mice [444]. In summary, several studies suggest that impaired FAAH function may confer a phenotype of high voluntary alcohol intake.

Acute alcohol withdrawal was associated with significant reductions in gene expression of several components of the ECS, including *FAAH, MGLL, CNR1, CNR2*, and *GPR55r* in the AMY. Although similar alterations in FAAH mRNA were evident following either continuous or intermittent alcohol exposure, alterations in MAGL, CB1r, CB2r, and GPR55r were more pronounced following intermittent exposure. These findings suggest that alcohol dependence and withdrawal are associated with dysregulated endocannabinoid signaling in the AMY [445]. Similarly, another study used a model of intermittent alcohol exposure (ethanol 20%, 3 g/kg injections for four days/week, for four weeks) of male Wistar rats and a subsequent alcohol deprivation in which these rats were assessed for the gene expression of different components of the ECS. Alcohol-exposed rats expressed higher and lower mRNA levels of endocannabinoid synthetic enzymes NAPE-PLD and DGLs in the medial PFC and the AMY, respectively. Furthermore, lower mRNA levels of CB1r, CB2r, and peroxisome proliferator-activated receptor-α (PPARα) were observed in the striatum [446] (Table 2).


ijms-23-04764-t002_Table 2Table 2Summary of main findings regarding the alterations of the ECS components in animals and human studies in Alcohol Use Disorders.Alcohol Use DisordersReferencesType of SampleType of EvaluationOutcomes[428]HumansNeuroimaging (PET)AUD showed ↑ CB1r binding in a circuit that included the AMY, HIPP, PT, insula, anterior and posterior cingulate cortices, and OFC.[433]HumansNeuroimaging (PET)AD subjects showed ↓ CB1r binding during early abstinence (3–7 days), which remained reduced during protracted abstinence (2–4 weeks).[429]HumansNeuroimaging (PET)Acute alcohol consumption resulted in a ↑ CB1r availabilityChronic alcohol drinking resulted in a ↓ CB1r availability that remained unaltered after abstinence (1 month).[437]Humans
*Post-mortem*
Cloninger type 1 alcohol dependent subjects showed ↑ DHEA levels in the AMY and a negative correlation between AEA concentrations and mGlu1/5 receptor density in the HIPP compared to Cloninger type 2 alcohol-dependent subjects and controls.[438]Humans
*Post-mortem*
CB1r protein expression in the PFC of the suicidal alcohol-dependent groupAlcohol-dependent subjects, regardless of the cause of death, ↓ MAGL activity, ↓ ERK, and ↓ CREB levels.[154]Humans
*Post-mortem*
Alcohol-dependent subjects presented hyper-functional CB1r in the caudate nucleus Non-suicidal alcohol-dependent subjects showed hypofunctional CB1r in the cerebellum.[439]AnimalsCB1R KO vs WT miceCB1r KO mice exhibited voluntary alcohol consumption and completely lacked alcohol-induced DA release in the NAcc compared to WT mice.[440]AnimalsCB1R KO vs WT miceCB1r KO mice displayed ↓ OH-induced CPP compared to WT mice. This ↓ OH-induced CPP exhibited by CB1r KO mice was correlated with an increase in striatum D2/D3 receptors.[441]AnimalsCB1r KO vs WT miceCB1r KO mice ↓OH consumption and preference, compared to WT miceCB1r KO mice were more sensitive to the acute alcohol effects than WT mice. The severity of alcohol withdrawal was also increased in CB1r KO mice[139,436]AnimalsC57/BJ6 male miceMice with high-alcohol preference had a lower gene expression of *CNR2* at the ventral midbrain[442]AnimalsCB2r KO vs WT miceCB2r KO mice presented ↑ a response to alcohol effects, OH-induced CPP, voluntary OH intake and preference, acquisition of alcohol self-administration, and motivation to drink alcohol compared to WT mice.[443]AnimalsFAAH gene KO vs WT miceFAAH KO mice showed a ↑ preference for alcohol and consumed more alcohol than WT miceThere were no significant differences between FAAH KO and WT mice in the severity of alcohol induced acute withdrawal, CPP, or sensitivity to the hypnotic effect of alcohol. FAAH KO mice showed a shorter duration and a faster recovery from intoxicating effects induced by alcohol.[444]AnimalsFAAH gene KO vs WT miceFemale FAAH KO mice had an ↑ alcohol intake and preference, were less sensitive to the effects of acute alcohol, and no CB1r levels and function down-regulation after voluntary alcohol consumption, compared to male FAAH KO, and male and female WT mice.[445]AnimalsMale Wistar rats exposed to continuous OH access vs intermittent OH accessAlcohol withdrawal was associated with significant ↓ mRNA expression FAAH, MAGL, CB1r, CB2r, and GPR55r in the AMY. ↓ MAGL, CB1r, CB2r, and GPR55r were more pronounced following intermittent alcohol exposure.[446]AnimalsMale Wistar rats exposed to intermittent OH accessAlcohol-exposed rats expressed ↑ mRNA levels of NAPE-PLD and DGL in the mPFC and the AMY, respectively, and ↓mRNA levels of CB1r, CB2r, and PPARα in the striatum.


### 10.3. Cannabis Use Disorders

#### 10.3.1. Clinical Studies

##### Neuroimaging Studies

Neuroimaging studies have examined associations between different components of the ECS and brain areas involved in other aspects of CaUDs. A study using PET and the [^18^F] FMPEP-d2 ligand in chronic daily cannabis smokers (*n* = 30) and control subjects with minimal lifetime exposure to cannabis (*n* = 28) evaluated CB1r binding. Assessments were performed immediately after chronic daily cannabis smoking and after four weeks of abstinence. CB1r downregulation correlated with years of cannabis smoking and was selective to cortical brain regions. However, after four weeks of continuously monitored abstinence from cannabis, CB1r density returned to normal levels [447]. These results were replicated in a further study that also determined brain areas affected by cannabis use. PET and the selective high-affinity ligand [(18) F]MK-9470 were used to obtain in vivo measurements of cerebral CB1r availability in chronic cannabis users (*n* = 10) compared to age-matched healthy subjects (*n* = 10). Each patient underwent a PET scan within the first week following the last cannabis consumption. Compared to the controls, cannabis users showed a global decrease in CB1r availability. This CB1r decrease was significant in the temporal lobe, the anterior and PCC, and the NAcc [448]. To assess the precise time course of changes in CB1r availability and down-regulation in cannabis-dependent subjects, a more recent study used High-Resolution Research Tomography and [^11^C] OMAR, in male cannabis-dependent subjects (*n* = 11) and matched healthy controls (*n* = 19). Cannabis-dependent subjects were scanned at baseline while actively using cannabis and after two days and 28 days of monitored abstinence, whereas healthy controls were examined at the baseline and 28 days later. Compared to healthy controls, [^11^C] OMAR *V*_T_ was lower in cannabis-dependent subjects at baseline in almost all brain regions. However, these group differences in CB1r availability were no longer evident after just two days of monitored abstinence from cannabis, and no significant group differences in CB1r availability were observed after 28 days of abstinence. There was a robust negative correlation between CB1r availability and withdrawal symptoms after two days of abstinence [449]. To sum up, all these studies suggest the existence of CB1r downregulation in active cannabis consumers, which begins to reverse upon the cessation of cannabis use (Table 3).

##### Genetic Studies

The most frequent genetic studies relating to the ECS and CaUDs explored the role of the *CNR1* gene and the *FAAH* gene in cannabis use, withdrawal, and relapse. Hopfer et al. (2006) [450] examined the association between the *CNR1* gene and cannabis dependence in 541 adolescents and young adults who used cannabis five or more times. They found that SNP rs806380, located in intron 2 of the *CNR1* gene, was significantly associated with developing cannabis dependence symptoms, with the G allele having a protective effect [450]. These results were subsequently replicated in other studies [451,452].

A study with 105 university students (18–25 years old) who reported smoking marijuana daily evaluated withdrawal symptoms and cue-elicited craving. Associations between withdrawal after abstinence and cue-elicited craving and polymorphisms of *CNR1* (rs2023239) and *FAAH* (rs324420) genes were explored. *CNR1* SNP displayed a significant abstinence x genotype interaction on withdrawal, while the FAAH SNP displayed a significant abstinence x genotype interaction on craving [453].

DNA samples and fMRI data using a cue-elicited craving paradigm were collected from 3-day-abstinent regular marijuana users (*n* = 37) in a subsequent study. Participants were grouped according to their genotype on two SNPs: rs2023239 in the *CNR1* gene and rs324420 in the *FAAH* gene. It was shown that carriers of the *CNR1* rs2023239 G allele had significantly greater activity in several brain regions during exposure to marijuana cues than those with the A/A genotype for this SNP.

The role of genes related to the *CNR2* has been less explored. A genetic study evaluating genotype distributions of gene variants of patients with synthetic cannabinoid use disorder (SCUD) compared to healthy controls found that *CNR2* rs2229579 variants were significantly different in patients diagnosed with SCUD compared with the control group [454] (Table 6).

##### *Post-Mortem* Studies

A study of cannabis smoke abusers that used [3H]SR141716A found a reduction in [3H]SR141716A binding in the HIPP compared to the brains of non-cannabis users. The density of CB1r mRNA was significantly lower in the brains of cannabis users for the caudate nucleus, PT, NAcc, and hippocampal region. These results suggest a down-regulation of the CB1r in several brain regions of chronic cannabis users [455].

A combined neuroimaging and *post-mortem* study evaluated white matter microstructural integrity, grey matter cortical thickness and density differences between cannabis-dependent subjects (*n* = 89) and matched controls (*n* = 89) and tested whether these cortical patterns were associated with the expression of genes relevant for cannabinoid signaling. Cannabis-dependent subjects presented lower fractional anisotropy in white matter and significantly less grey matter than the controls. In addition, spatial patterns of grey matter differences were significantly associated with regional differences in MAGL expression in *post-mortem* brain tissue. These results reflect that the regions with high MAGL expression may be more vulnerable to the effects of chronic cannabis use on cortical thickness [456] (Table 3).

#### 10.3.2. Animal Studies

Preclinical studies suggest that chronic cannabis exposure is linked to decreased CB1r expression and CB1r down-regulation/desensitization. A time course of 21 days of continuous exposure to THC revealed increased CB1r gene (*CNR1*) expression in the cerebellum and HIPP and reduced in the striatum until day 14. However, *CNR1* expression in all three brain areas returned to control levels by day 21 of THC treatment once behavioral tolerance had been developed [457]. The contents of both AEA and 2-AG, CB1r and [35S] GTPγS binding were determined in several brain regions. THC-tolerant rats decreased CB1r and [35S] GTPγS binding in most brain areas except the limbic forebrain. In addition, AEA increased in the limbic forebrain, and AEA and 2-AG decreased in the striatum. These results show that prolonged activation of CB1r leads to decreased endocannabinoid contents and signaling in the striatum and to increased AEA formation in the limbic forebrain [458].

Long-term depression in VTA GABA neurons is dependent on the CB1r, and it is involved in the rewarding effects of drugs of abuse. Indeed, it was found that long-term depression of VTA GABA neurons was absent in CB1r KO but preserved in WT mice. In addition, THC also produced a long-term depression in the WT but not in CB1r KO mice [459] (Table 3).


ijms-23-04764-t003_Table 3Table 3Summary of main findings regarding the alterations of the ECS components in animals and human studies in Cannabis Use Disorders.Cannabis Use DisordersReferencesType of SampleType of EvaluationOutcomes[447]HumansNeuroimaging (PET)CB1r downregulation in years of THC smokersAfter 4 weeks of abstinence, CB1r density returned to normal levels.[293]HumansNeuroimaging (PET)THC users showed an ↓ in CB1r availability, significant in the temporal lobe, the anterior and PCC, and in the NAcc. [449]HumansNeuroimaging (HRRT)THC-dependent subjects showed ↓ CB1r availability Differences in CB1r availability were no longer evident after 2 days of abstinence, and no significant group differences in CB1r availability after 28 days of abstinence.[455]HumanPost-mortemIn chronic cannabis users, CB1r binding was ↓ in the HIPP, caudate nucleus, PT, and NAcc. [456]HumanPost-mortemIn THC-dependent subjects, regions with higher MAGL expression are more vulnerable to cortical thinning.[457]AnimalRats exposed to THCCB1r mRNA levels were increased in the cerebellum and HIPP and reduced in the striatum until day 14. CB1r expression in all three brain areas returned to control levels by day 21 of THC treatment once behavioral tolerance had been developed. [458]AnimalsTHC-tolerant ratsTHC-tolerant rats exhibited an ↓in CB1r and [35S]GTPγS binding in most brain areas, except the limbic forebrain. AEA ↑ in the limbic forebrain, and AEA and 2-AG ↓in the striatum.[459]AnimalsCB1r KO vs WT miceLong-term depression of VTA GABA neurons was absent in CB1r KO but preserved in WT mice. THC produced a long-term depression in the WT but not in CB1r KO mice. 


### 10.4. Cocaine and Other Stimulant Use Disorders

#### 10.4.1. Clinical Studies

##### Genetic Studies

A study with four large samples of European and African Americans investigated the association of *CNR1* gene SNPs with cocaine use disorders (CoUDs) and cocaine-induced paranoia. Two independent *CNR1* SNPs, rs6454674 and rs806368, significantly increased the risk for CoUD. In addition, rs806368 showed a significant association with cocaine-induced paranoia [460]. A more recent study investigated the association between two SNPs in the *CNR1* gene (rs6454674, rs806368) and CoUD in African Americans to replicate these results. A significant difference in genotype frequencies was found between subjects with CoUD and healthy controls for both SNP (Table 6).

The SNP rs324420 of the *FAAH* gene, related to decreased FAAH activity and increased endocannabinoid potentiation, was explored in CoUD participants (*n* = 70). Participants rated 10 subjective effect measures using a Visual Analog Scale before and following cocaine administration. It was found that participants with the SNPs rs324420 of the *FAAH* gene showed an increased cocaine-induced subjective rating [461] (Table 6).

##### Post-Mortem Studies

A study comparing mixed cocaine-opiate addicts, opiate addicts, and their respective matched controls found that in cocaine addicts, but not in mixed cocaine-opiate or opiate abusers, CB1r and GRK2/3/5 were reduced in the PFC, whereas CB2r was not significantly altered. Therefore, the dysregulation of CB1r and GRK2/3/5 signaling by cocaine may alter neuroplasticity in the brains of cocaine addicts [462] (Table 4).

#### 10.4.2. Animal Studies

Pioneering research used the self-administration paradigm to evaluate the role of CB1r in several aspects of cocaine reward including acquisition, maintenance, and motivation to seek the drug. CB1r KO and WT littermate mice were trained to self-administer cocaine. Only 25% of the CB1r KO mice, compared to the 75% of their WT littermates, acquired a reliable operant responding to the self-administration of cocaine. This study demonstrated that the CB1r is essential in consolidating cocaine reinforcement [463].

Further research examined whether CB1r regulated cocaine-seeking in glutamatergic and GABAergic neurons. CB1r expression in forebrain GABAergic neurons-controlled sensitivity to cocaine, while CB1r expression in cortical glutamatergic neurons controlled the associative learning processes. These findings demonstrate an altered balance of glutamatergic-CB1r and GABA-CB1r activity that could participate in the vulnerability to cocaine abuse and addiction [464]. A recent study assessed the performance of cocaine locomotor and associative behavioral assays in D1-*CNR1* KO mice and adenosine (A2a)-*CNR1* KO mice. The results suggest that CB1r plays an essential role in cocaine-induced locomotor activation and are necessary for cocaine reward perception, memory consolidation, and recall [465].

A study assessing the involvement of the CB1r of the lateral habenula on the performance in the 5-choice serial reaction time task (5CSRTT) in male Long–Evans rats found that systemic cocaine increased premature responding [466].

Several studies have highlighted the functional importance of CB2r in the actions produced by psychostimulant drugs such as cocaine. Our group used transgenic mice overexpressing the CB2r and WT mice in experimental paradigms to evaluate the motor (acute and sensitized locomotor responses to cocaine), reinforcing in the CPP paradigm, and motivational (cocaine intravenous self-administration) effects of cocaine. Overexpression of the CB2r significantly decreased motor response to acute administration of cocaine, cocaine-induced motor sensitization, CPP, and cocaine self-administration [467].

*DAT-CNR2* KO mice showed enhanced psychostimulant-induced hyperactivity and lack of psychostimulant-induced sensitization. The deletion of CB2r in DA neurons modified TH levels and reduced DAT gene expression in midbrain regions in the DAT-*CNR2* KO mice [429].

Several studies have compared the role of CB1r and CB2r in different components of CoUDs. A study that used an intravenous cocaine self-administration paradigm analyzed the effects of the selective CB2r agonist JWH133 in CB1r and CB2r KO and WT mice. WT and CB1r KO mice, but not CB2r KO mice, exhibited cocaine-enhanced locomotion and NAcc extracellular DA. These studies suggest that CB1r could be involved in cocaine-reinforced behaviors, whereas CB2r may be associated with cocaine-evoked adaptation [468] (Table 4).


ijms-23-04764-t004_Table 4Table 4Summary of the main findings regarding the alterations of ECS components in animals and human studies in Cocaine Use Disorders.Cocaine Use DisordersReferencesType of SampleType of EvaluationOutcomes[462]Human
*Post-mortem*
↓ CB1r and GRK2/3/5 in the PFC in CoUDs[463]AnimalsCB1r KO vs WT mice25% of the CB1r KO mice compared to the 75% of their WT littermates acquired a reliable operant responding to self-administration of cocaine, and the number of sessions required to attain this behavior was ↑ in CB1r KO mice.[464]AnimalsGlu-CB1r vs GABA-CB1r KO vs WTCB1r expression in forebrain GABAergic neurons-controlled sensitivity to cocaine, while CB1r expression in cortical glutamatergic neurons controlled the associative learning processes.[465]Animals*D1-CNR1* KO vs *A2a-CNR1* KO vs WT mice*D1-CNR1* KO mice did not display hyperlocomotion in response to acute cocaine dosing. *D1-CNR1* and *A2a-CNR1* KO mice exhibited blunted locomotor activity across repeated cocaine doses *A2a-CNR1* KO mice did not express a preference for cocaine paired environments in a two-choice place preference task.[466]AnimalsMale Long Evans rats Systemic cocaine increased premature responding, a measure of impulsivity.[467]AnimalsTransgenic mice overexpressing the CB2r vs WT littermatesOverexpression of the CB2r significantly ↓ motor response to acute administration of cocaine cocaine-induced motor sensitization, CPP, and cocaine self-administration.[427]Animals*DAT-CNR2* KO vs WT mice*DAT-CNR2* KO mice enhanced psychostimulant-induced hyperactivity but an absence of psychostimulant-induced sensitization compared to WT mice.[468]AnimalsMale Wistar ratsFollowing cocaine self-administration, a ↑ CB1r expression in the VTA and a ↓ CB1r expression in the PFC, dorsal striatum, and AMY. Cocaine abstinence, ↑CB1r expression in the SN and the AMY, and a ↓ CB2r expression in the PFC, NAcc, and medial globus pallidus.


### 10.5. Opiate Use Disorders

#### 10.5.1. Clinical Studies

##### Plasma Studies

One published study evaluated the expression of CB2r in PBMCs in morphine abusers (*n* = 8) compared to healthy controls (*n* = 5). It was found that CB2r was upregulated in the PBMCs of morphine abusers [469] (Table 5).

##### Genetic Studies

A recent study assessed associations between genetic polymorphisms and methadone maintenance therapy dosing [470]. The association between the SNP rs2023239 of the *CNR1* gene and lifetime MDD and suicidal behavior was examined in a population of opiate-dependent outpatients remitted under stable methadone treatment. The results revealed that within Caucasian stabilized, methadone-maintained outpatients, the minor C allele of the SNP rs2023239 of the *CNR1* gene was associated with a lower prevalence of MDD but not with changes in the history of attempted suicide [232] (Table 6).

#### 10.5.2. Animal Studies

The involvement of CB1r in the motivational properties and rewarding effects of opiates and the development of physical dependence on opiates suggests an interconnection between CB1r and opiate receptors in brain areas that mediate addictive behaviors. Indeed, the reinforcing properties of morphine and the severity of the morphine withdrawal syndrome were strongly reduced in CB1r KO compared to WT mice [471]. In another study, morphine did not induce intravenous self-administration in CB1r KO mice, unlike WT mice, who significantly self-administered morphine.

The involvement of CB1r in morphine and cocaine motivational effects in CB1r KO mice was examined using a CPP and the sensitization to the locomotor responses induced by these drugs. The hyperlocomotion induced by acute morphine administration was preserved, whereas the sensitization to this locomotor response induced by chronic morphine treatment was abolished in CB1r KO mice. These results add further evidence that the CB1r is essential for adaptive responses produced by chronic morphine but not by chronic cocaine treatment [472].

CB2r was upregulated in the spleen and PBMCs [469] in Sprague-Dawley rats under morphine exposure compared to the controls. In addition, a more recent study compared CB2rKO and WT mice using LY2828360, a slow signaling G protein-biased cannabinoid CB2r agonist, and morphine. In WT mice, LY2828360 blocked morphine-induced reward in a CPP paradigm, whereas the LY2828360-induced attenuation of morphine-induced reward was absent in CB2r KO mice. These results suggest that CB2r activation attenuates opiate reward and dependence [473].

The role of AEA and 2-AG has also been explored. Maternally deprived adolescent rats exhibited higher concentrations of AEA than adolescent non-deprived rats in the NAcc, the CPu nucleus, and the mesencephalon. In contrast, maternally deprived adult rats increased AEA and 2-AG levels in the NAcc and 2-AG in the CPu nucleus compared to non-deprived adult rats. This study suggests that altered brain endocannabinoid levels may contribute to the escalation behavior in the morphine consumption test [474] (Table 5).


ijms-23-04764-t005_Table 5Table 5Summary of main findings regarding the alterations of the ECS components in animals and human studies in Cocaine Use Disorders.Opiate Use DisordersAuthorsType of SampleType of EvaluationOutcomes[469]HumansPeripheralPlasmaIn morphine abusers, CB2r were upregulated in the PBMCs.[471]AnimalsCB1r KO vs WT miceCB1r KO mice, the reinforcing properties of morphine and the severity of the morphine withdrawal syndrome were strongly ↓.[472]AnimalsCB1r KO vs WT miceThe sensitization to the locomotor response induced by chronic morphine treatment was abolished in CB1r KO mice.Morphine induced a CPP in WT mice but failed to produce any response in CB1r KO mice[469]AnimalsSprague-Dawley rats under morphine exposure vs control ratsRats under morphine exposure exhibited CB2r upregulation in the spleen and PBMCs[473]AnimalsCBr2 KO vs KO miceIn WT mice, LY2828360 blocked morphine-induced reward in a CPP paradigm, whereas morphine-induced reward was absent in CB2r KO mice. LY2828360 partially attenuated naloxone-precipitated opioid withdrawal in morphine-dependent WT mice, whereas this withdrawal was markedly exacerbated in CB2r KO mice[474]AnimalsMaternally deprived adolescent ratsMaternally deprived adolescent rats exhibited ↑ AEA in the NAcc, the Cpu nucleus, and the mesencephalonMaternally deprived adult rats, showed ↑ AEA and 2-AG in the NAcc, and ↑ 2-AG in the CPu nucleus, 
ijms-23-04764-t006_Table 6Table 6Summary of the main ECS candidate genes polymorphisms associated with each disorder.GeneSNPDisorderAuthors
*CNR1*
rs110402rs7209436Crs242924Grs7766029rs1049353rs2180619rs806366rs806367rs806368rs806369rs806370rs806371rs806379rs806380rs2023239rs6454674rs1049353rs12720071rs1535255AnxietyAnxietyAnxietyAnxiety, SchizophreniaPTSD, Depression, Schizophrenia, ADHD, EDPTSDDepression, SchizophreniaDepressionDepression, ADHD, ED, OUD, AUD, CaUDs, CoUDsEDDepressionDepressionNUD, AUDCaUDsDepression, Schizophrenia, ADHD, NUD, OUD, AUD, CaUDsDepression, AUD, CoUDsSchizophrenia, ADHD, AUDSchizophreniaADHD, AUD[172,174][174][174][177,273,275][193,195,196,230,231,233,269,271,271,276,375,376,396,397,430,434][134][234,271,396][234][234,269,270,375,396,430,434,451,460,470][396][234][157,230][269,376,419,435][450,451][133,232,274,276,375,419,432,433,435,452,453][234,430,434,460][193,195,196,230,231,233,269,272,277,375,376,396,397,430,434][273,275][277,279,419,435]
*CNR2*
rs2501432Depression, Schizophrenia[139,233,299,300]rs12744386Schizophrenia[135,299]rs35761398Schizophrenia[135]Q63RAUD[299,436]rs2229579CaUds[454]
*FAAH*
rs324420PTSD, Depression, Schizophrenia, CaUDs, CoUDs[95,172,173,174,175,181,196,310,434,453,461]rs2295633ADHD[377]385 A/A genotypeED[407]ADHD: Attention Deficit Hyperactivity disorder; AUD: Alcohol Use Disorder; CaUDs: Cannabis Use Disorder; ED: Eating Disorder; CoUDs: Cocaine Use Disorder; NUD: Nicotine Use Disorder; OUD: Opioid Use Disorder; PTSD: post-Traumatic stress disorder.


## 11. Concluding Remarks

The involvement of the ECS in the etiology and functional neurochemistry of psychiatric and neurodevelopment disorders is undeniable. There is an urgent need to develop new therapeutic strategies that, alone or in combination with the drugs currently used in psychiatry, may improve the efficacy and safety of psychiatric disorder treatment. In this respect, the modulation of the ECS has emerged as a significant opportunity to treat psychiatric diseases. Even though more clinical trials are necessary, the evidence included in this review provides an overview of the opportunities that cannabinoid receptors, endogenous cannabinoid ligands, or the drugs regulating their metabolism may offer as potential therapeutic targets in clinical psychiatry. In each section of this review, we were able to see how the ECS components are altered and the role they play in the modulation and development of different psychiatric disorders, considering that this system is also critical in the different stages of neurodevelopment and the normal development of the CNS. In the last 50 years, much progress has been made in translational research on cannabinoids, but there is still a long way before cannabinoid drugs can be commercialized and used to treat these disorders. It is essential to highlight the methods used to detect molecular alterations of the ECS, such as neuroimaging techniques (PET, fMRI) and blood samples (PBMCs, plasma) to isolate of eCBs ligands, genetic, epigenetic, and proteomics modifications to identify new biomarkers and therapeutic targets for the treatment of psychiatric diseases. Animal studies are also critical in determining the alterations in ECS omponents and designing new double-blind, PLB-controlled clinical trials with cannabinoids compounds that may be potentially beneficial for the patient. 

To sum up, more clinical research studies are urgently needed for two different purposes: (1) To identify alterations of the ECS components that are potentially useful as new biomarkers relating to a specific disease or condition, which may result in markers of status, trait, or therapeutic evolution, and (2) to determine the molecular alterations in the ECS, facilitating the design of new therapeutic targets based on the specific alterations found to improve pharmacological treatment. Thus, it is essential to note that greater effort is required to design and perform more clinical studies, significantly increasing the sample sizes to achieve greater reproducibility and developing new therapeutic drugs to incorporate into clinical practice.

## 12. Methods

The literature review consisted of an exhaustive search for scientific information in the Medline database (PubMed). The following keywords and their combinations were used for the “methods to identify alterations in the endocannabinoid system section”: “Endocannabinoids” AND “chemistry techniques, analytical”; “Chemistry techniques, analytical.” For the sections, “Anxiety-related disorders”, “depression”, “schizophrenia”, “ASD”, “ADHD”, “eating disorders”, and “addiction disorders” the following keywords and their combinations were used: anxiety OR stress AND cannabinoid AND human OR animal, post-traumatic stress disorder AND human OR animal; “schizophrenia” and “cannabinoid” were combined with terms related with the technical approximation by the Boolean operator “AND”; “depression” and “cannabinoid” were combined with terms related to the technical approximation by the Boolean operator “AND”; “ASD”, “ADHD”, “eating disorders” and “cannabinoid” were combined with terms related to the technical approximation by the Boolean operator “AND”; “nicotine use disorders” AND cannabinoids AND humans OR animals; “alcohol use disorders” AND “cannabinoids” AND humans OR animals; “cannabis use disorders” AND “cannabinoids” AND humans OR animals; “cocaine use disorders” AND cannabinoids AND humans OR animals; “opioid use disorders” AND “cannabinoids” AND humans OR animals. Additional searches included references from identified publications. In the screening process, articles published in predatory journals and studies published earlier than 1980 were excluded.

## Figures and Tables

**Figure 1 ijms-23-04764-f001:**
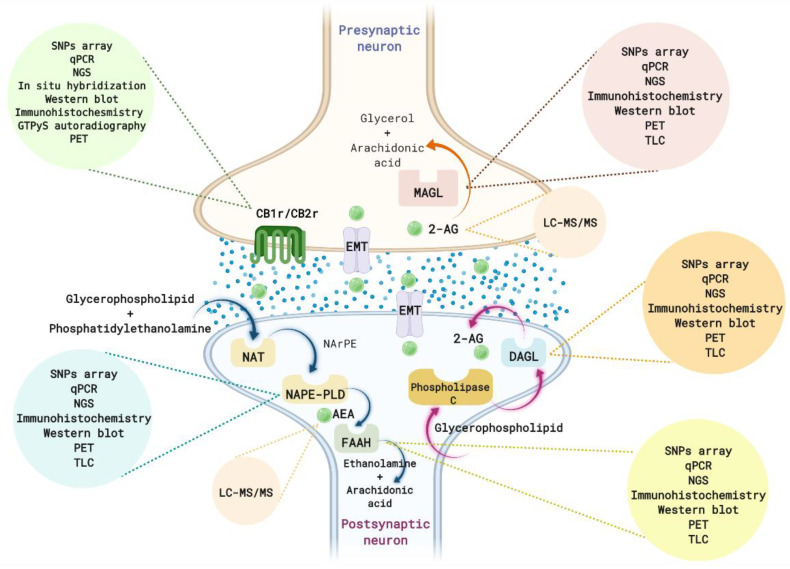
Schematic representation of the main methods used for identifying alterations in the ECS. AEA: anandamide; 2-AG: 2-arachidonoylglycerol; CB1r: cannabinoid 1 receptor; CB2r: cannabinoid 2 receptor; DAGL: diacylglycerol lipase; EMT: endocannabinoid membrane transporter; FAAH: fatty acid amide hydrolase; GTPyS: GTPgammaS or guanosine 5′-O-(γ-thio) triphosphate; LC-MS / MS: liquid chromatography-tandem mass spectrometry; MAGL: monoacylglycerol lipase; NarPE: N-arachidonoyl phosphatidylethanolamine; NAPE-PLD: N-acylphosphatidylethanolamine specific phospholipase D; NAT: N-acyl-transferase; NGS: next-generation sequencing (genomics); PET: positron emission tomography; q-PCR: real-time polymerase chain reaction; SNPs: single nucleotide polymorphisms; TLC: thin-layer chromatography.

**Figure 2 ijms-23-04764-f002:**
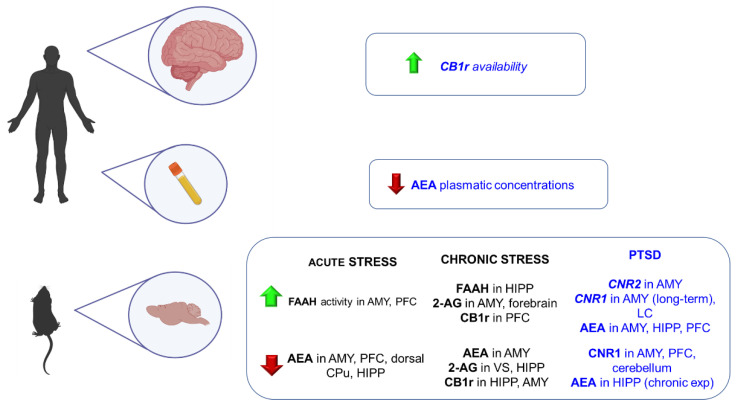
Main findings regarding the alterations of the ECS components in animal models and patients with anxiety and post-traumatic stress disorder. 2-AG: 2-arachidonoylglycerol; AEA: anandamide; AMY: amygdala; CB1r: cannabinoid receptor 1; *CNR1*: gene encoding CB1r; *CNR2*: gene encoding CB2r; CPu: caudate putamen; FAAH: fatty acid amide hydrolase; HIPP: hippocampus; PFC: prefrontal cortex; VS: ventral striatum. Note: Anxiety-related studies are in black, and post-traumatic stress disorder studies are in blue.

**Figure 3 ijms-23-04764-f003:**
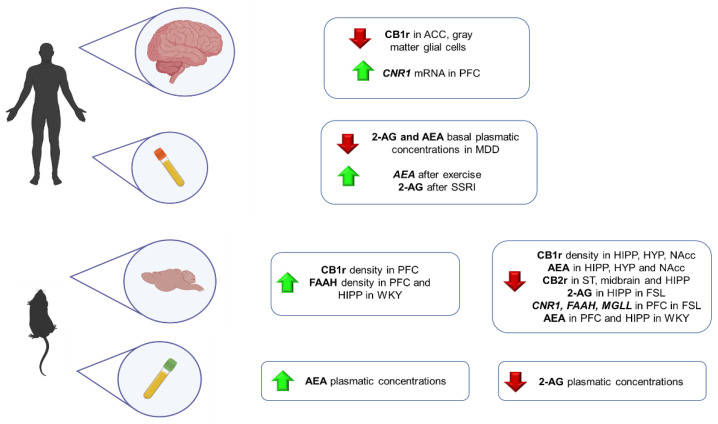
Main findings regarding the alterations of ECS components in animal models and patients with depression. 2-AG: 2-arachidonoylglycerol; ACC: anterior cingulate cortex; AEA: anandamide; CB1r: cannabinoid receptor 1; *CNR1*: gene encoding CB1r; *FAAH:* gene encoding fatty acid amide hydrolase; FSL: Flinders Sensitive Line rats; HIPP: hippocampus; HYP: hypothalamus; MDD: major depressive disorder; MGLL: gene encoding MAGL; NAcc: nucleus accumbens; PFC: prefrontal cortex; SSRI: selective serotonin reuptake inhibitors; ST: striatum; WKY: Wistar Kyoto rats.

**Figure 4 ijms-23-04764-f004:**
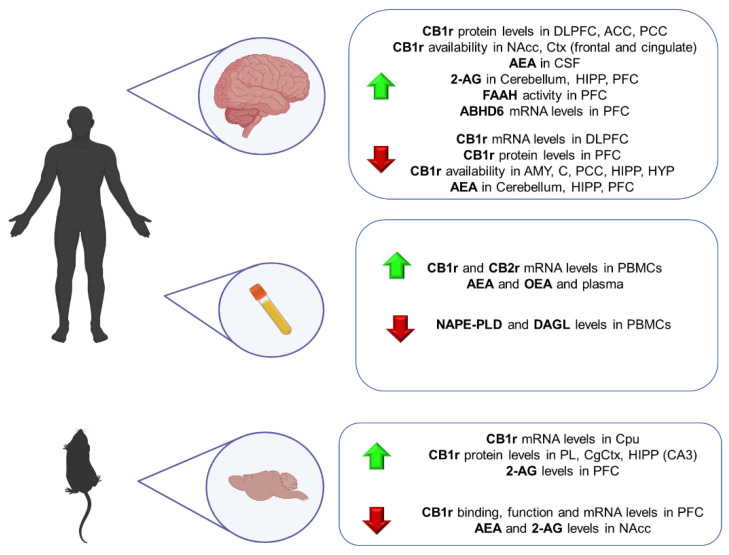
Main findings regarding the alterations of the ECS components in animal models and patients with schizophrenia. 2-AG: 2-arachidonoylglycerol; ACC: anterior cingulate cortex; AEA: anandamide; AMY: amygdala; CB1r: cannabinoid receptor 1; CB2r: cannabinoid receptor 2; Ctx: cortex; CSF: cerebrospinal fluid; DAGL: diacylglycerol lipase; DLPFC: dorsolateral prefrontal cortex; FAAH: fatty acid amide hydrolase; HIPP: hippocampus; HIP: hypothalamus; MAGL: monoacylglycerol lipase; NAcc: nucleus accumbens; NAPE-PLD: N-acyl phosphatidylethanolamine phospholipase; OEA: ethanolamide; PCC: posterior cingulate cortex; PFC: prefrontal cortex.

**Figure 5 ijms-23-04764-f005:**
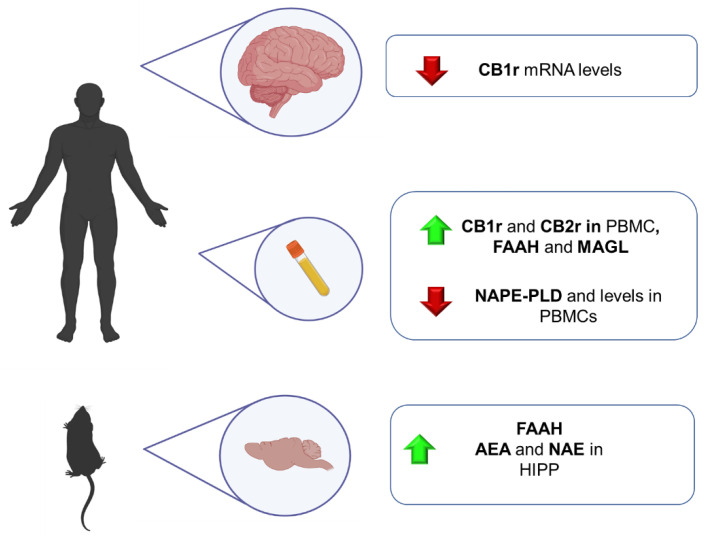
Main findings regarding the alterations of the ECS components in animal models and patients with autism spectrum disorders. AEA: anandamide; CB1r: cannabinoid receptor 1; CB2r: cannabinoid receptor 2; FAAH: fatty acid amide hydrolase; HIPP: hippocampus; NAPE-PLD: N-acyl phosphatidylethanolamine phospholipase, NAE: N- acetyl ethanolamine.

**Figure 6 ijms-23-04764-f006:**
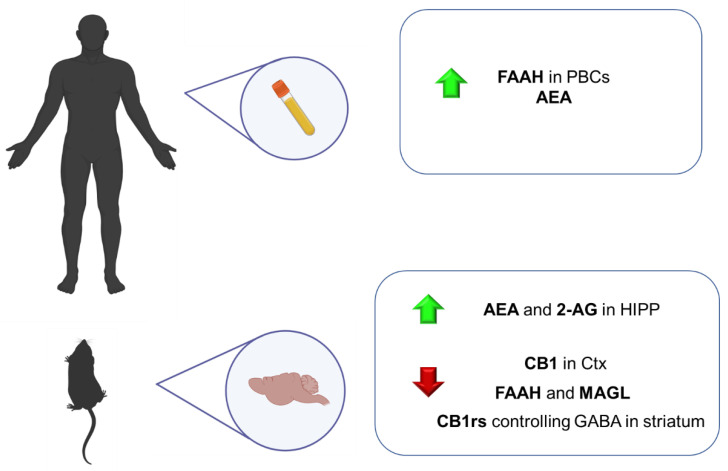
Main findings regarding the alterations of the ECS components in animal models and patients with attention deficit hyperactivity disorder. AEA: anandamide; CB1r: cannabinoid receptor 1; FAAH: fatty acid amide hydrolase; HIPP: hippocampus; Ctx: cortex.

**Figure 7 ijms-23-04764-f007:**
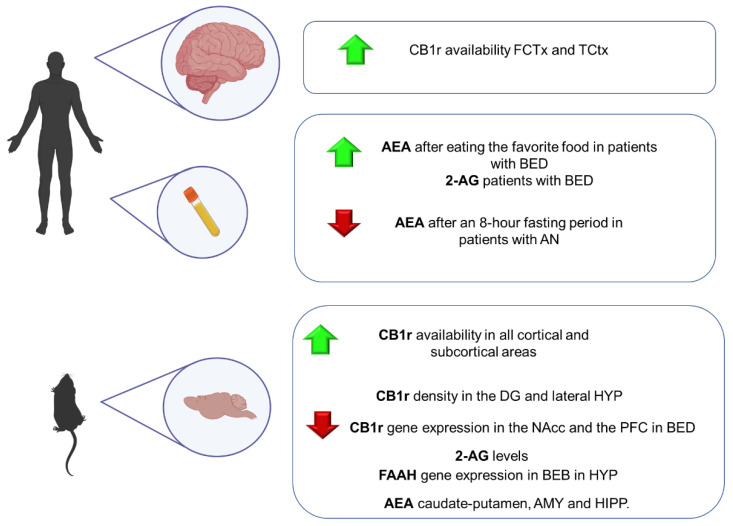
Main findings regarding the alterations of the ECS components in animal models and patients with eating disorders. 2-AG: 2-arachidonoylglycerol; AEA: anandamide; CB1r: cannabinoid receptor 1; *CNR1*: gene encoding CB1r; HIPP: hippocampus; HYP: hypothalamus; BED: binge eating disorders; AN: Anorexia nervosa; DG: dentate gyrus; Nacc: nucleus accumbens; PFC: prefrontal cortex; FCTx: frontal cortex, TCtx: temporal cortex; BEB: binge eating behavior.

## Data Availability

Not applicable.

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
