# Peer review of "Molecular Alterations of the Endocannabinoid System in Psychiatric Disorders"

_ijms, 2022, doi:10.3390/ijms23094764_

Round 1
Reviewer 1 Report
The authors analyzed in a rigorous and detailed way the molecular alterations of the endocannabinoid system in psychiatric disorders. The authors discussed a number of both clinical and animal studies of notable significance. The article is written very clearly.
Author Response
Many thanks for the comments. We appreciate your opinion about our work.

Reviewer 2 Report
In their work, the authors of the manuscript entitled "Molecular alterations of the endocannabinoid system in psychiatric disorders" offer a comprehensive review of the literature. They reviewed how molecular alterations of the endocannabinoid system (ECS) can modulate different CNS functions with the development of various neuropsychiatric disorders. Undoubtedly, so far there are a lot of qualitative reviews regarding the connection between ECS and these disorders. However, this work is very extensive, well-structured, documented in an exceptional way by the impressive number of bibliographical references (467). The large number of tables and images makes the material easy to read and understand. Each disorder has been supported with both preclinical and clinical studies, emphasizing the need to extend clinical trials in order to develop new therapeutic strategies, with a modulatory role on the ECS.
I have only a few minor comments:
- On page 9, figure 1: not all acronyms are explained in the legend (eg. MAGL, GTPyS, LC-MS / MS, NarPE etc). Also, CB1r appears in the legend but in the image appears without letter r. Likewise, CB2 appears in the legend without letter r. It would be advisable to have a uniform expression throughout the text.
- At the end, the last line should be deleted (line 2920).
Author Response
"Please see the attachment."

Reviewer 3 Report
The purpose of this review is to collect relevant literature on the role of the endocannabinoid system in the neurobiology of mental illness. The authors collected and critically reviewed a large amount of information (more than 460 references) on the changes in the endocannabinoid system constituents that accompany psychiatric and neurodevelopment disorders. In general, the review makes it possible to assess the undoubted potential of a therapeutic effect on the elements of the endocannabinoid system to increase the effectiveness of psychiatric care.
Minor remarks.
2-AG and its metabolite 1-AG line 174 correctly - 2-AG and its isomer 1-AG
In the section 2 (A brief overview of the ECS) the attention was concentrated on two main cannabinoid receptors. However ECS include some so-called non-classical CB-receptors too. For example, GPR55 is referred as no-CB1/non-CB2 receptor and it plays significant role in various pathologies. Authors provide information on possible involvement of this receptor in suicide cases, lines 782-789 (please, place relevant reference at the end of this paragraph). It is recommended to update the section 2 of review with data on mentioned receptors and give an explanation why authors exclude this receptor family from consideration in the present manuscript.
In the section 3.2.2 line 228 the citation of recent review: The epigenetics of the endocannabinoid system, Meccariello et al, doi: 10.3390/ijms21031113 was omitted.
Author Response
"Please see the attachment."
